# Contextualizing the think crisis-think female stereotype in explaining the glass cliff: Gendered traits, gender, and type of crisis

**Clara Kulich**[1]*, **Leire Gartzia**[2], **Meera Komarraju**[3], **Cristina Aelenei**[4]

**1** Faculty of Psychology and Educational Sciences, University of Geneva, Geneva, Switzerland, **2** Deusto Business School, University of Deusto, Deusto, Spain, **3** Department of Psychology, Southern Illinois University Carbondale, Carbondale, Illinois, United States of America, **4** Institute of Psychology, Université de Paris, Paris, France

* clara.kulich@unige.ch

**Data Availability Statement:** The data underlying this study are available on OSF (https://doi.org/10.17605/OSF.IO/B3U5K).

## Abstract

The glass cliff suggests that women are more likely to access leadership positions when organizations are facing a crisis. Although this phenomenon is well established, it is still largely unknown how variations in types of crises influence the strength of the think crisis-think female association, and whether female leaders and leaders with communal gendered traits are both affected by this association. We hypothesized that selection of stereotypically feminine traits (communal leaders) is specific to a relational crisis because of a fit between leader traits and traits required by the situation. We further expected that the selection of women also extends to other crisis situations because other factors such as their signaling change potential may play a role. We investigated the associations that participants made with candidates who varied across gendered traits and gender and between two crisis situations involving problems with either stereotypically feminine (e.g., an internal disharmony) or masculine (e.g., a financial problem) components, and a no crisis situation control. Results from three experimental studies ($N$s = 319, 384, 385) supported our hypotheses by showing that communal leaders were most strongly associated with a relational crisis and least with a financial crisis, with the no crisis context situated in-between. This pattern was explained by higher relevance ratings for communal leadership behavior in the relational crisis versus financial crisis context, with the no crisis context situated in-between. In contrast, female leaders were most strongly associated with the relational crisis and least with the no crisis context, with the financial crisis situated in-between. Specific explanatory mechanisms related to the female-crisis association are explored and discussed. Our findings suggest that implicit motivations for valuing feminine/communal leadership and atypical leaders in crisis situations need further research.

**Funding:** CK received grant no. 100019_188934 from the Swiss National Science Foundation http://www.snf.ch/en/Pages/default.aspx. The funder had no role in study design, data collection and analysis, decision to publish, or preparation of the manuscript.

**Competing interests:** The authors have declared that no competing interests exist.

# Introduction

Although more women are attaining managerial roles in organizations, they continue to be underrepresented in top management positions [1]. Explanations for this disparity include the "think manager–think male" paradigm [2], which refers to the perceived incongruity between the characteristics typically ascribed to women and the characteristics typically ascribed to leaders [3]. In recent years, the potential "advantage of female leadership" has been discussed in modern organizations requiring more transformational and relationship focused leadership styles [4, 5]. However, academics have warned that simply associating feminine typed leadership with an increased preference for female leaders, as such, ignores the "importance of contextual contingencies" [6] (p655), see also [7]. More recent analyses of leadership in contexts of crisis do indeed suggest increased leadership opportunities for women in organizations suffering some sort of crisis, such as poor performance or scandals—situations where success is relatively improbable [8–10], a phenomenon, commonly known as the "glass cliff" [11]. The "glass" metaphor refers to the subtle discriminatory nature of this phenomenon as precarious appointments are more likely for minority groups. Moreover, these subtleties can obscure the reality of discrimination for a specific individual, as the phenomenon only becomes visible by considering several cases in aggregate. The "cliff" metaphor also relates to the fact that precarious positions likely expose occupants to more scrutiny and criticism, and higher levels of stress, in addition to the risk of failure, with serious consequences for their careers [12].

Research on explanations for the glass cliff has elicited interest in the context-dependence of leader ideals, as studies indicate that the male-manager stereotype is less consistent and may deviate from the male prototype in certain crisis contexts, such as poor organizational performance [13]. The purpose of the present research is to extend this field of study by investigating the nuances of different types of crises, their impact on variations in preference for "feminine", or communal leadership traits in crisis management, and to further test whether these preferences extend to the "female" gender which may be associated with feminine traits through stereotypes. In particular, we investigated the gendered nature of different types of crises involving relational elements (e.g., internal disharmony) versus financial elements (e.g., inappropriate financial decision making), expecting that varied crisis typologies elicit distinct perceptions of which leader characteristics would be effective in a crisis. We experimentally investigated when and why different crisis types were associated with preferences for communal traits versus female gender in leaders.

## The glass cliff

Variations in the leadership context have been shown to have an impact on perceptions of the effectiveness of female and male leadership [3] in a meta-analysis [4]. The idea that effective leadership emerges dynamically and that situational factors are critical in understanding leadership effectiveness has been advanced by a variety of leadership theories, for a review see [14–18]. Yet, this literature has not provided a comprehensive understanding of the interplay of gendered contextual demands and perceived leadership effectiveness in adverse scenarios [11].

A crisis is generally a difficult situation that threatens relevant organizational goals and typically calls for a quick response. In certain crisis contexts, the survival of an organization is explicitly linked to relational dimensions that define leadership effectiveness in terms of positive interpersonal relationships and group performance [19–22]. According to social role theory these communal dimensions are associated with stereotypically feminine roles, and therefore with women, cf. [23, 24]. In other crisis contexts, goal orientation and the ability of leaders to respond to immediate threats in a direct and even authoritarian way are perceived as relevant for the survival of an organization [25–27]. These agentic dimensions are associated

with stereotypically masculine functions, and therefore men [23, 28]. Thus, based on social role theory the tendency to select women as effective leaders in crisis situations should be influenced by the gendered nature of the crisis [13].

Some investigations of the glass cliff in leadership have narrowed in on the preference of women (as a gender) in certain crisis situations. Other scholars have concentrated on the link between "feminine" or communal traits and crisis contexts. In the following, we will discuss these two lines, gender and gendered traits, separately.

## The glass cliff and the preference for women

Consistent with the idea that certain crisis contexts require a relevant number of stereotypically feminine or communal behaviors, a growing number of studies have shown that in these contexts the manager-male association is reduced. This is the case in organizations experiencing difficult conditions where a "think crisis-think female" stereotype has been found to be more applicable [10, 29, 30].

## Gender stereotypes

Some of the studies have shown that women have higher chances to be selected as leaders in a crisis situation compared to a non-crisis context, and other studies have shown that women are preferred over men during crises [11]. In these studies experimenters have typically presented similar CVs of male and female candidates, with gender being the only information differing between them. This has led to the proposition that women are more likely to be chosen due to gender stereotypes based on which they are presumed to possess the communal traits deemed useful in a crisis context. Indeed, some evidence suggests that stereotypically feminine attributes such as shared leadership, teamwork, and emotional management can be relevant dimensions of effective leadership in crisis situations that are characterized by a decrease in a company's profits [19, 20, 31]. In many crisis situations, a leader's ability to adopt a perspective and to influence team members in order to obtain cooperation and collaboration in achieving common goals is critical in enhancing long-term corporate performance and sustainability [32]. Followers who trust their leaders are also more likely to sustain focus and effort towards achieving organizational goals, particularly when facing extended periods of stress [22]. In contrast, a lack of trust in the integrity of a leader's decisions can diminish cohesion and commitment even during relatively short crises [33]. Moreover, research on perceptions of leaders in a crisis situation has shown that women who behave in relational ways may be allocated higher trust than male managers, but only if the technical solution to the problem is known [34]. This suggests that women displaying communal behavior may indeed have a leadership advantage over men, but only if a crisis explicitly requires stereotypically feminine leadership competences, but not more stereotypically masculine competences, to solve the crisis. This supports the claim [35] that an acute crisis often calls for autocratic management, and when this point is solved the relational dimension becomes of higher importance.

Following this idea, an experiment [13] (Study 1) investigated whether the male-manager association holds in a crisis situation, and demonstrated that while this link holds in a healthy company context, people associated feminine traits with leadership in a crisis context. Thus, an underlying motivation for the choice of a woman may be that her gender is associated with communal leadership deemed more effective in the given situation. The choice of a woman could thus be in the "best interest" of the ailing company's functioning, with the chosen woman being expected to effectively change the situation due to her competencies.

## Signaling change

Other scholars have investigated the strategic choice of women for their atypicality. The occurrence of a crisis leads to more public attention to the organization concerned and the image repair strategies put in place need to be well designed, as they will impact an organization's overall reputation [36]. One such strategy may be to deviate from the male leader prototype by choosing a woman, using the gender contrast as a visible sign of change [37]. The motive for female leadership assignments may then be to symbolize change [29, 38]. In support of this hypothesis, an experimental study [37] showed that women were not chosen for their leadership qualifications, but rather were selected as a way of signaling change to clients, investors, and customers.

Overall, the research on preferences for female gendered leaders in times of crisis shows that women may be chosen due to the stereotypical attributions made (i.e., communal traits and behavior) based on their gender, but they may also be chosen for their potential to signal change. The latter case may occur as an isolated explanatory process [37], but perceptions of the special competences of women and their direct impact on a company's functioning could also act in concert with perceptions that a woman is perceived as a signal of change. For example, recent research has shown that choices of ethnic minority individuals for hard-to-win seats in political races may be associated with both actual change (competence based) and signaling change motivations [39].

## The glass cliff and the preference for feminine traits

Research on the glass cliff has also investigated the effects of gendered traits. For instance, experimental research [13] has shown that the perceived suitability of stereotypically masculine versus feminine traits in times of crisis is contingent upon what is explicitly required from the manager. In particular, when a stereotypically feminine management role (e.g., people management), or a non-agentic role (e.g., taking the blame for the crisis) was required, feminine rather than masculine traits were perceived as more desirable, thus suggesting a think crisis-think female phenomenon. To capture these potential differences between selection of women and selection of communal traits in crisis contexts, previous research has called for studies clearly differentiating between two components of gender in line with a "think crisis–think in a stereotypically feminine way association" [30].

This research [13] revealed a key observation that a preference for stereotypically feminine traits did not occur if more agentic tasks, such as being an active spokesperson for the company or performance improvement was required. In fact, crisis contexts that are associated with stereotypically masculine dimensions such as financial, competitive, or technological problems are particularly likely to be linked to centralized, autocratic leadership. For instance, when organizations face extremely critical events, leadership that is directive and transactional is featured as being most effective [26, 35]. Further, there is experimental [40] and field evidence based on stereotypically masculine settings, for example, on an aircraft carrier [27], indicating that leaders who exercise power by being directive and goal-oriented are more effective during extreme events. Leaders who provide rapid and authoritative responses are more likely to be followed in such contexts regardless of the nature of their decisions [41]. Although relationship-oriented leadership behaviors are sometimes implemented in advanced stages of a crisis, in the first phase, authoritarian expressions of leadership typically occur [35]. Followers are also more likely to accept autocratic leadership (stereotypically masculine) in threatening situations that are poorly defined [25]. Research has also shown that stereotypically feminine, communal qualities are perceived as a hindrance to performance in task-oriented managerial activities such as managing a financial transaction, improving manufacturing processes, or increasing profits [42].

The research we have discussed has mostly examined gendered traits in isolation. As exception, one study [30] examined selection of both women and communal traits for a crisis context in scenarios with different leadership referents (agentic vs. communal) and sexism scores. Findings showed that selection of communal female leaders was generally higher in situations with communal referents and low sexism, but explicit variations of crises were not included. To our knowledge only one experimental work has investigated gendered traits simultaneously with information on the gender of candidates themselves. These experiments [43] manipulated both a candidate's gender and their gendered traits, revealing that agentic (versus communal) leadership traits were preferred when leadership appointments were focused on choosing a person effective and actively engaged in improving company performance. Candidate gender did not systematically affect these choices.

Overall, we can conclude that gender and gendered traits cannot be interchangeably used as they lead to different effects [44]. The organizational context, whether it is a crisis or not, and what type of expectations are held of a new leader, play an important role. Two shortcomings should be considered in relation to prior research. First, gender was not investigated in the studies by [13], thus it is not clear whether it is a think crisis-think "feminine" phenomenon, or whether a leader of female gender is preferred because a woman is assumed to have feminine traits. And second, while previous studies [43] investigated both gendered trait information and the gender of candidates, they did not vary the type of crisis.

## Gendered traits, gender, and crisis type

Following the findings outlined in the prior section, a crucial remaining question is whether or not the evidence, that communal traits are preferred in some crisis types and agentic traits in others, can be directly related to the choice of leaders in terms of their gender, and whether or not the mechanisms leading to the choice of a leader due to female gender or due to stereotypically feminine traits differ.

We therefore aimed to extend the research on the think crisis-think female stereotype by investigating how generalizable it is across crisis typologies while measuring preferences for both agentic versus communal and male versus female leaders. In so doing, we also extend theory with regard to glass cliff research. First, we overcome the gap identified in previous think crisis-think female studies which did not address the effects of gendered traits and gender separately [30] by manipulating candidate profiles accordingly. Second, we varied the gendered nature of the problem (relational or financial) and measured their association with preferences for distinct candidate profiles. Finally, in prior research [13] the gender dimension was examined by evaluating reactions to the traits and competences that were presented as a requirement for different crisis situations (i.e., situations that required being able to "manage people and personnel issues", or "take control of the division and improve performance"; [13] p478). However, such an approach provided explicit information about the required skills in a crisis instead of focusing on people's own expectations. We avoided this conflation by asking people to infer from a crisis type which traits and behaviors were deemed relevant in order to better understand the mental models underlying different crisis scenarios.

## Hypotheses

**Leader preferences.** Here we outline our reasoning and expectations for the choice of communal and female leaders separately, and then provide a rationale for the expected mechanisms. We argue that when a crisis type explicitly involves communal problems (e.g., disharmony between employees), communal leaders should be preferred, whereas problems of a financial nature should trigger a need for the default agentic leader. The default no crisis

managerial condition, however, has largely been shown to trigger think manager-think male associations. Thus, such a "neutral" situation, without additional specifications of the nature of the tasks to be handled, should tend towards the default male or agentic leader, but we expected to a lesser degree compared to a financial crisis, which implicitly demands strong agentic leading. Previous research [43] has also demonstrated that the choice of agentic leaders is stronger in a financial crisis compared to a no crisis situation. We thus predict in *Hypothesis 1*: A communal (as opposed to agentic) leader will be preferred in a crisis context that contains communal problems (e.g., relational disharmony between employees) over a crisis scenario that contains agentic problems (e.g., financial problems), and a no crisis context will be situated in-between.

The choice of a female leader may be motivated by gender stereotypes which associate communal traits with female gender [13]. Thus, female leaders should be more likely to be chosen in a crisis highlighting relational problems because women are associated with strengths in interpersonal relationships [45]. In contrast, when a crisis type explicitly involves problems typically associated with agentic and male competences (e.g., financial problems), the think crisis-think female stereotype should decrease in strength. However, female gender can also be used to symbolize a change from traditional male leadership in crisis situations [37]. Such a motivation may affect any crisis type. Therefore, the selection of a female leader in a financial crisis should be situated in-between a relational and a no crisis situation. Overall, the highest likelihood for a woman to be chosen should be in a relational crisis, where women are valued because they are stereotyped as having the "communal" competencies required to manage the crisis effectively, and at the same time, respond to other motivational needs, such as signaling change to outside actors. In the financial crisis context, signaling change motivations would remain, provoking a female choice, but the added importance of "communal" leadership traits would be diminished, as communal traits are less valued in the financial crisis context. Finally, the no crisis context should show the lowest likelihood for a woman to emerge as a leader, following the think manager-think male principle. In *Hypothesis 2*, which concerns gender, we thus also predicted a progressive pattern, however of a different shape than for gendered traits: A female leader (as opposed to a male) will be preferred in a crisis context that contains communal problems compared to a no crisis condition, and a crisis scenario that contains agentic problems will be situated in-between.

**Mechanism.** So far, we have argued that relational crisis contexts may elicit communal and female leadership preferences because communal behavior and traits may be seen as valued competences in such a context. We anticipate that the relevance attributed to relational qualities will be higher in a crisis with communal problems than in a no-crisis situation, or when compared to a crisis showing financial problems, which leads to *Hypothesis 3*: Participants will ascribe higher relevance to communal leadership (i.e., behavior and traits) in a relational crisis scenario compared to a financial crisis, with the no-crisis context situated in-between. We expect this outcome to account for the progressive preference effect of communal candidates predicted in H1.

We did not make inverse predictions for agency, as the backlash literature on agentic female candidates shows that even if agency is considered relevant, higher relevance of agency does not necessarily lead to the choice of agentic women [46]. In the context of glass cliff choices, we argue that the presence of communion-related attributions is likely to be decisive for the preference of communal leaders, rather than the absence of agentic ones.

We did not predict a mediational pattern for the choice of women, as this choice may also depend on a number of other factors, such as a woman's perceived "atypicality" in a managerial role, roles which are generally male-dominated [9, 37].

## The present studies

In order to test our hypotheses, we created crisis situations that clearly invoked either stereo-typically masculine (a competitive market environment combined with inappropriate financial decisions) or stereotypically feminine elements (internal disharmony which seriously damaged employees' relations and motivations). As the glass cliff is defined as the choice of female leaders in a crisis as compared to a no-crisis condition, cf. [47], we added a control condition in which no crisis was presented. The following three studies examined two gendered crisis types (relational versus financial) in comparison to a no crisis context. Studies 1 and 2 were structured as a 3 (Crisis Type: relational, financial, no crisis) between-participants design measuring participant choice between four candidates who varied across gendered traits (agentic versus communal) and gender (male versus female). Study 3 consisted of a replication of the results of Studies 1 and 2 with an inverted experimental design by presenting a 4 (Candidate: agentic male, agentic female, communal male, communal female) between-participants design measuring the participant choice between the three organizational contexts (relational versus financial versus no crisis).

## Organizational role

Reasoning for our hypotheses is based on the assumption that decision makers in our studies take the perspective of the organization and want to make decisions that help it succeed. People, however, hold different schemas about responsibilities and relational dependencies depending on the position or role they occupy within an organization, see [48]. This is likely the case for managers or others who are explicitly asked to meet organizational goals. A leader's responsibilities are often associated with organizational systems and policies, which place them in the position of having to more closely align their actions to the decisions and guidelines established by the organization [49, 50]. In contrast, followers are typically the beneficiaries, or alternatively, the victims of a leader's actions [51]. Along these lines, one should expect that an individual's organizational position (being in the role of a leader versus an employee) would have an effect on their mental representations of what effective leadership should look like in a crisis situation. Leader-primed decision-makers should more strongly emphasize strategic oriented leadership and be more sensitive to the organizational needs of the specific situation (e.g., crisis or no crisis). In an effort to find the optimal solution for a given context, leaders might more strongly engage in the motivations described above. In contrast, employee-primed decision-makers should be more sensitive to their own needs and thus seek a relationship focused leader regardless of the context. As the recruitment for leadership positions is usually done by decision makers with managing functions in companies, rather than employees, the angle of the decision maker perspective is more likely to be source of glass cliffs decisions. Moreover, the potential motivations for glass cliff decisions discussed here also take the perspective of those in charge of hiring. Thus, in the present research we asked participants to focus primarily on this role.

In most glass cliff studies, participants are in fact placed in the position of imagining their role as a manager or recruiter making decisions about a new CEO appointment, an approach we followed in Studies 2 and 3, which included samples of workers, by asking them explicitly to think about organizational goals in their decision making. By contrast, Study 1 used a student sample. As student jobs do not offer the full perspective of organizational dynamics, and they are unlikely to occupy leadership positions, we manipulated in this first study the type of perspective they were to take: as an employee or as a leader, expecting that the effects in Hypotheses 1 and 2 would more likely, or more strongly, occur for participants in the role of a leader than in the role of an employee.

## Data transparency

The data that support our findings for all studies are openly available in OSF at https://osf.io/
b3u5k/?view_only=1be338fdb1534ca48f121f95b3fa9679. All manipulations, and all exclusions
of the three reported studies are reported in the manuscript. All measures are mentioned in
the manuscript and described in detail in the S1 File. We performed data analysis with the
sample sizes provided herein. No additional data were sought for any of the studies after initial
data-analysis. Participants in all studies were randomly assigned to experimental conditions.

## Study 1

The major aim of this study was to test Hypotheses 1, 2, and 3, and to explore the impact of the
organizational role (employee versus leader) adopted by participants.

### Method

Study 1 was conducted in Spain where no ethic approval was demanded by the university.
Study design and procedure, however, followed standard ethic guidelines for research on
humans. Participants gave informed consent at the beginning of the study by clicking on a
button.

**Participants.** Participants were 319 business administration students (53.4% men) in
Spain who participated in the study in exchange for course credit. We excluded from the origi-
nal sample ($N$ = 324) participants who had not made a candidate choice ($n$ = 3), and who did
not indicate their gender ($n$ = 2). Participants reported their age ($M$ = 19.87, $SD$ = 1.52) and
previous work experience whereby 64.9% of the participants indicated they had at least one
year of previous work experience outside the university ($M$ = 11.82 months, $SD$ = 14.16).

We conducted effect-size sensitivity analyses for the effects of crisis type (the C1 contrasts)
in a logistic regression on preferences for candidate gendered traits and preferences for candi-
date gender, using G$^*$Power 3 [52]. With the Study 1 sample size (N = 319), $\alpha$ = .05, and 80%
desired power, the minimum effect size that we could detect is an Odd Ratio = 1.89 (based on
a probability under H0, p1 = 0.5).

**Procedure.** After consenting to participate, we asked participants to think about their pro-
fessional future in an organization following a *possible selves* procedure [53, 54]. Participants
envisioned themselves as making a decision about the most appropriate candidate for a ficti-
tious organization. The description of the organization (see Appendix A1a and A1b Table in
S1 Appendix) incorporated a contextual variation of a dramatic decrease in the company's
profits, with the origin of this crisis either described as *financial* ("the company lost out to the
competition" and "the financial forecasts have not been adequate"), or as *relational* ("harmony
problem had seriously damaged the motivation of employees and created a negative atmo-
sphere"). In addition, a *no-crisis* control condition was presented where no information about
the company's performance was given. To capture the potential effects of a participant's role
expectations in the organization, we included an additional condition that manipulated expec-
tations about their specific role in the organization. In the *leader role* scenario, we asked partic-
ipants to imagine themselves holding a position of responsibility in the company. In the
*employee role* condition, participants were asked to imagine themselves working for the com-
pany. This resulted in a 3 (Crisis Type: relational crisis, financial crisis, no crisis) × 2 (Organi-
zational Role: leader versus employee) between-participants design.

After reading the organizational scenario, participants were asked to evaluate the relevance
of communion and agency (leadership behaviors and traits) for the position, and then to eval-
uate a list of five job candidates (leader suitability) and to select the most suitable candidate to
ensure appropriate organizational functioning in the company.

**Candidate profiles.** Following previous research on the glass cliff [9, 30], the description of the candidates consisted of a brief CV with a short biographical sketch of the candidates. Candidate gender was manipulated by using typically male or female first names taken from research using Spanish names [30]. Gender traits were manipulated in a similar manner to the approach taken by [43]. One male and one female candidate were described with agentic traits (e.g., self-confident, independent, decisive), and one of each gender was described with communal traits (e.g., other-oriented, considerate, kind). To avoid overlap between words, traits were described with different adjectives that have a similar meaning and derived from previous studies [55]. Exact profiles can be found in Appendix A2 Table in S1 Appendix.

To make the manipulation less obvious [9] and consistent with the prevalence of male and stereotypically masculine leaders in organizations [28], we provided a more realistic list of candidates (more men than women) by including an additional male candidate. He was described with extreme, unmitigated agentic traits that included negative content (e.g., authoritarian, tough, competitive; see [56]), aiming to disqualify him by this unsympathetic description.

**Measures.** Response scales ranged from 1 *Not at all relevant* to 6 *Extremely relevant*. Additional measures and explorative analyses for these can be found in the S1 File (i.e., leader suitability, relevance ratings of agentic behavior and agentic traits and filler items, characteristics of selected candidate).

*Communal leadership attributes.* Models in the leadership literature consider both leadership traits and behaviors [17]. Traits are generally considered personality elements that are inherited or acquired through socialization and are often connected to gendered traits of identity that are viewed as more stable across different situations, traits such as being sensitive, empathetic, or kind. In contrast, behaviors or styles of leadership (e.g., transformational-transactional) are considered to be learned, assuming that one can be trained to flexibly use them across different situations (for a review comparing leadership effects across these dimensions see [57]). Because stereotypes concern expectations for how people are (or should be) and how they behave (or should) [58], we measured both traits and behaviors in an effort to capture a global picture of communal and agentic aspects of leadership. We present here only the communal dimension which concerns our H3. The agentic dimension can be found in the S2 Table in the S1 File.

Participants indicated the extent to which a set of leadership behaviors taken from the Competing Values Management Practices Survey [59] was relevant for the candidate in the manipulated scenario [56]. We used four people-related items from the "mentor" dimension which can be generally subsumed under the communion dimension (Listen to the personal problems of subordinates; Show empathy and concern in dealing with subordinates; Treat each individual in a sensitive, caring way; Show concern for the needs of subordinates, $\alpha = .83$; $M = 4.75$, $SD = 0.86$).

Participants further indicated the extent to which a set of eight communal traits taken from the Personal Attributes Questionnaire (PAQ) [60] were relevant for the candidate in the manipulated scenario (e.g., *being devoted to others*, *emotional*, *attentive*, *understanding*, *kind*, *helpful*, *warm*, *aware of others' feelings*, $\alpha = .86$; $M = 4.54$, $SD = 0.74$).

## Results

**Preparatory analyses.** We first looked at general preferences for the five candidates. As expected, the extreme agentic candidate was chosen least (7.2% versus 31.7% agentic male, 22.6% agentic female, 22.6% communal female, and 16.3% communal male).

For the analyses reported below, we controlled whether a participant's gender affected the results. We added it as a main effect and in all interactions. The main effects of crisis type

reported below remained. Additional effects which included participant gender occurred only for the leadership behavior and traits analyses which we report in the S1 File.

**Preferences for a candidate's gendered traits.** We conducted a logistic regression on the choice of candidates as a function of gendered traits (0 = agentic candidates, 1 = communal candidates). Following our prediction in H1, we entered two orthogonal contrasts for crisis type (Contrast 1: 1 = relational crisis, 0 = no crisis, -1 = financial crisis; Contrast 2: 1 = relational and financial crises, -2 = no crisis), and organizational role (-1 = leader; 1 = employee) as well as their interactions. The C1 contrast tests the difference between relational crisis versus financial crisis. The C2 contrast verifies whether the no crisis condition is situated in-between by testing the comparison between the no crisis versus relational and financial crises taken together. To support our hypothesis, contrast C1 should be significant, but not C2. Consistent with our main hypothesis H1, the C1 contrast showed that overall participants were more likely to select communal leaders in the relational crisis (50.00%) than in the financial crisis (27.20%), $B = 0.49$, $\chi^2$ (1, N = 319) = 12.15, $p = .001$, $e^B = 1.64$, 95% CI [1.24, 2.16] (Table 1, $e^B$ represents the odds ratio), with the no crisis context (38.50%) situated in-between (i.e., contrast C2 had no effect, $B = 0.02$, $\chi^2$ (1, N = 319) = 0.03, $p = .861$, $e^B = 1.02$, 95% CI [0.85, 1.21]). Overall, agentic candidates were chosen more than communal candidates, $B = -0.51$, $\chi^2$ (1, N = 319) = 17.91, $p < .001$, $e^B = 0.60$. No further main or interaction effects occurred ($p$s > .118).

**Preferences for candidate gender.** We conducted a logistic regression on the choice of candidate gender (0 = male candidates, 1 = female candidates). Following our prediction in H2 where choices would be most frequent in the relational crisis, and least frequent in the control condition, with the financial crisis situated in-between, we entered two orthogonal contrasts for crisis type (Contrast 1: 1 = relational crisis, 0 = financial crisis, -1 = no crisis; Contrast 2: 1 = relational crisis and no crisis, -2 = financial crisis), and organizational role (-1 = leader, 1 = employee) as well as their interactions. The effect of crisis type C1 that was expected in H2, $B = 0.31$, $\chi^2$ (1, N = 319) = 4.52, $p = .034$, $e^B = 1.36$, 95% CI [1.02, 1.81], demonstrated that overall participants were more likely to select women when exposed to the relational crisis (51.8%) than when exposed to the no-crisis condition (37.4%), whereas the financial crisis (43.9%) was situated in-between (i.e., the residual contrast C2 was not significant, $B = -0.001$, $\chi^2$ (1, N = 319) < 0.001, $p = .995$, $e^B = 1.00$, 95% CI [0.86, 1.17]. Table 1 summarizes these results. Overall, male candidates were chosen more than female candidates, $B = -0.24$, $\chi^2$ (1, N = 319) = 4.26, $p = .039$, $e^B = 0.79$. No further main or interaction effects occurred ($p$s > .316).

**Perceived relevance of leadership attributes.** For the test of H3, we were interested in the importance of relevance ratings for communal leadership for choices in terms of candidate's gendered traits (communal versus agentic). In a first step, we present two separate ANOVAs for relevance ratings for communal behavior and communal traits as dependent variables. The

**Table 1. Choice of a communal versus agentic and female versus male candidates as a function of type of crisis (Study 1, Spain).**

|  | Choice of candidates % (n) | | | |
|---|---|---|---|---|
|  | **Relational crisis** | **Financial crisis** | **No crisis** | **Total** |
| Communal candidates | 50.0 (57) | 27.2 (31) | 38.5 (35) | 38.6 (123) |
| Agentic candidates | 50.0 (57) | 72.8 (83) | 61.5 (56) | 61.4 (196) |
| Total | 100.0 (114) | 100.0 (114) | 100.0 (91) | 100.0 (319) |
| Female candidates | 51.8 (59) | 43.9 (50) | 37.4 (34) | 44.8 (143) |
| Male candidates | 48.2 (55) | 56.1 (64) | 62.6 (57) | 55.2 (176) |
| Total | 100.0 (114) | 100.0 (114) | 100.0 (91) | 100.0 (319) |

C1 contrast of crisis type (1 = relational crisis, 0 = no crisis, -1 = financial crisis), the C2 residual contrast (1 = relational crisis and financial crisis, -2 = no crisis), and organizational role (-1 = leader, 1 = employee) were entered as independent variables, as well as interactions between the contrasts and organizational role.

For communal leadership behavior, the C1 contrast was significant, $F(1, 313) = 7.87$, $p = .005$, $\eta_p^2 = .03$, showing that communal behavior was evaluated as more relevant in the relational crisis ($M = 4.87$, $SE = 0.08$) than in the financial crisis ($M = 4.56$, $SE = 0.08$), with the no crisis context ($M = 4.85$, $SE = 0.09$) situated in-between (C2 was not significant, $F(1, 313) = 1.54$, $p = .215$, $\eta_p^2 = .005$). Organizational role also had an effect, $F(1, 313) = 7.87$, $p = .005$, $\eta_p^2 = .03$, in the sense that participants in the leader role ($M = 4.66$, $SE = 0.07$) judged communal traits as less important than those in the employee role ($M = 4.86$, $SE = 0.07$). Interactions with role were not significant ($p > .313$).

For communal leadership traits, only the contrast C2 was significant, $F(1,313) = 4.30$, $p = .039$, $\eta_p^2 = .01$ (C1: $F(1,313) = 0.30$, $p = .588$, $\eta_p^2 = .001$), indicating that communal traits were perceived as more important in a no crisis context ($M = 4.68$, $SE = 0.08$) when compared to the two crisis situations combined (relational: $M = 4.46$, $SE = 0.07$; financial: $M = 4.51$, $SE = 0.07$). No other effects occurred ($ps > .140$).

**Mediational analyses.** In order to fully test whether higher ratings of the relevance for communal behavior could explain the preferential choice of the communal leader in the relational crisis, as compared to the no crisis, and the financial crisis conditions (Hypothesis 3), we conducted a mediation model 4 using Hayes' [61] PROCESS macro with 10,000 biased bootstrap samples. PROCESS can estimate models with a binary outcome. The dependent measure was candidate gendered traits (0 = agentic, 1 = communal). The C1 contrast of crisis type (1 = relational crisis, 0 = no crisis, -1 = financial crisis) was entered as the independent variable, while controlling for C2 (1 = relational and financial crisis, -2 = no crisis), organizational role, and their interactions. Only communal leadership behavior was entered as mediator (see Fig 1), as the ANOVAs presented above revealed that the non-significant path 'a' for communal traits disqualified it as potential mediator.

For communal leadership behavior, path 'a' showed a positive incremental effect of financial crisis—no crisis—relational crisis (C1) on communal behavior, $B = 0.16$, $SE = 0.06$, $p = .005$, 95% CI [0.05, 0.27]. Path 'b' showed that the more communal behavior was rated as relevant, the more likely a communal candidate would be to be chosen, $B = 0.58$, $SE = 0.15$, $p < .001$, 95% CI [0.28, 0.88]. The direct effect of C1 on candidate selection was significant, path 'c': $B = 0.44$, $SE = 0.15$, $p = .003$, 95% CI [0.15, 0.72]. As we expected, communal behavior mediated the effect of crisis type C1 on the selection of communal candidates, $B = 0.09$, $SE = 0.04$, 95% CI [0.02, 0.20].

Thus, H3 was supported for communal leadership behavior, but not for communal leadership traits. We are confident that causality of this mediational model can be assumed for two

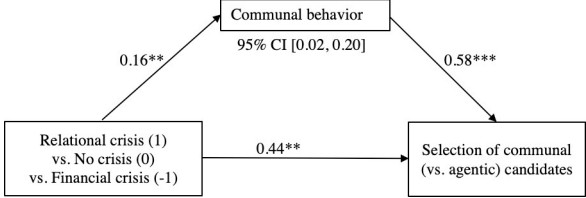

**Fig 1. Indirect effect of crisis type (contrast C1), mediated by communal leadership behavior, on the preference for communal (as opposed to agentic) candidates (Study 1, Spain).** Process Model 4 of Hayes (2018, [62]). Numbers indicate unstandardized coefficients. * $p < .05$, ** $p < .01$, *** $p < .001$.

reasons. First, the evaluation of the situation in terms of the relevance of communality (the mediator) was measured after reading about the organizational context (manipulated), but before the selection of the candidate (outcome). Second, the organizational context reported a problem, and the mediator measured a participant's perception of what kind of leadership behavior would be relevant to a leader whose mission it was to handle the problem. Following this evaluation participants were presented a list of candidates from which they could choose. The choice of the candidate according to their characteristics was likely aligned with the type of leadership behavior that was considered most relevant in the given context. Testing the reverse, the impact of candidate choice on the behaviors deemed relevant has no theoretical validity.

## Discussion

In support of the first hypothesis, communal leaders were preferred most in a relational, and least in a financial crisis situation, with the no crises scenario situated in between. These findings replicate and extend prior research [43], which demonstrated the leadership advantage of agentic (versus communal) candidates in a financial crisis versus a no crisis context, by showing that leaders with communal traits are most preferred when an organizational crisis is described as resulting from relational problems, thus requiring relational leadership. Moreover, consistent with our second hypothesis, women were selected most often in a relational, then in a financial, and least in a no-crisis context. This suggests that a candidate's female gender is linked to various kinds of crisis contexts, whereas a candidate's communal gendered traits are not, instead clearly linking to the relational–and not the financial—crisis. In this way, although preferences for the communal and female candidates were similar for the relational (50.0% versus 51.8%) and no crisis (38.5% versus 37.4%) contexts, communal candidates were clearly less favored in the financial context (27.2%), whereas the choices of female candidates (43.9%) in the financial context were more frequent and situated in-between the relational crisis and no crisis contexts. Regarding our exploratory analysis of organizational role, no effect on the preference variables was shown but we cannot conclude that this variable does not have an impact, as our study did not have the power to test this effect decisively.

Alternative mechanisms are likely at play with regard to gendered characteristics versus gender as a category. In fact, and in line with H3, a leader's communal behavior was perceived as more relevant in a relational crisis (than in a no crisis or financial crisis), and this explained the stronger preference of communal leaders in the relational context. This suggests that participants focused on a communal leader's match with behavioral expectations for a leader as a remedy for a relational crisis. Unexpectedly, the type of crisis did not significantly affect the ratings of relevance for communal traits. Thus, H3 was not supported for the trait dimension. The reasons for this inconclusive result could be a lack of power. However, a theoretical explanation may be that a relational crisis could be solved by anyone who uses the right style of leadership, i.e., communal behavior. Learned communal behaviors may be perceived by participants as more flexible, capable of being exhibited by anyone, be they female or male when it is relevant, as in a relational crisis. In contrast, communal leadership traits may be attributed by participants to internal characteristics and considered as more stable, and therefore potentially more strongly associated to women than men, with a chosen leader less agile to adapt to changing situations.

Conversely, the selection of female candidates in a relational crisis context, was neither explained by ratings of the relevance of leadership behavior (see S1 File). Previous research has shown that crisis contexts more generally favor females as leadership choice as a means to signal change [37], whereas preferences for gendered leadership traits are more closely linked to a leader's actual qualifications and capabilities [43]. Such divergent results suggest that candidate

gender and gendered trait dimensions have related but not interchangeable effects on candidate selection procedures. Our present research suggests that participants focused on a candidate's match with behavioral expectations (communal behavior) as a remedy for a relational crisis. Thus, communal behavior may be judged to be a learned "competence" to actually change the situation. Yet, communal leadership expectations may be only weakly or partially related to the choice of a female candidate. Female gender may instead be considered a visible sign of the implementation of a change, aiming to repair an organization's image in a crisis, independent of the gendered role demands based on crisis type.

## Study 2

The main aim of this study was to replicate and extend the findings of Study 1 with a larger sample size based on a priori power analysis. Moreover, to increase power, we used a simpler design by fixing the participant organizational role to that of a leader (therefore not manipulating organizational role), and we provided only a list of four candidates to choose from (the fifth extreme agentic male candidate was not included). In addition, we used a more context relevant sample, workers. Finally, we added explorative measures for the evaluation of a candidate's signaling potential, as this mechanism was not explored in Study 1, and may help to better understand the effects of candidate gender.

### Method

Ethic approval was obtained from the first author's faculty ethic comity for Study 2 (Faculty of Psychology and Educational Science of the University of Geneva, N˚ PSE.20190305.06). Participants gave written informed consent by clicking on one of two possible answers (Yes or No) at the beginning of the study just after reading general information about study content and data management, and again at the end of the questionnaire after having been fully debriefed about the experimental manipulations and aims of the study.

**Participants.** We aimed to recruit a sample size that would allow us to have 80% power to capture a minimum effect size of an Odd Ratio = 1.80 (with a probability under H0, p1 = .50). We specified a small effect size (Odd Ratio = 1.80) based on the published results in glass cliff literature employing similar experimental protocols [37, 43], and as suggested by results obtained in Study 1 for the main effects of the experimental condition. A G*Power 3 analysis [52] indicated that we needed approximately 375 participants. We requested 450 participants on MTurk. The final sample was $N = 384$ MTurk workers after exclusion of 60 participants from the original sample ($N = 444$ complete responses). A sensitivity analysis showed that with $N = 384$, $\alpha = .05$, and 80% desired power, the minimum detectable effect size was an Odd Ratio = 1.78 (based on a probability under H0, p1 = 0.5). Exclusion criteria were the following: Participants who did not give informed consent after the debriefing at the end of the experiment ($n = 9$), who did not pass the attention check ($n = 21$), who had missing data for measures used in the analyses ($n = 1$), who indicated a gender other than male or female ($n = 4$), who indicated they were an MTurk worker as a profession ($n = 1$), who were not English native speakers ($n = 16$), and the second trial of those who participated twice due to technical error (MTurk Worker IDs occurring twice, $n = 10$).

Participants included 52.9% women and reflected the ethnic composition of the USA (55.5% White, 25.5% Hispanic/Latino, 10.2% African American, 8.9% Asian American, 1.8% Native American). Participants were on average $M = 38.07$ years old ($SD = 11.81$; minimum 18, maximum 75 years). Participants had on average $M = 6.36$ years of work-experience with their current employer ($SD = 5.77$, minimum 1 year, maximum 44 years) and $M = 14.26$ years of work-experience overall ($SD = 10.82$, minimum 1 year, maximum 51 years); 88% were

employed (8% self-employed and the rest unemployed, in education, or retired); 84.4% worked full-time, and 42.7% held a management position.

**Procedure.** This study had a 3 (Crisis Type: relational crisis, financial crisis, no crisis) between-participant design with choice of candidate (male agentic, female agentic, male communal, female communal) as an outcome variable.

After consenting to participate, participants read a fictitious article about a company, "Jefferson", that is seeking a new CEO (inspired by [37, 43], see Appendix A1 Fig in S1 Appendix). Company performance was presented as either poor or as strong (hereafter between brackets). Poor performance was presented for the two crisis conditions and strong performance for the no crisis condition. The article was entitled "Going down . . . - Jefferson's disastrous performance" (versus "From Strength to Strength–Jefferson's outstanding performance") and the text read ". . . it has experienced a steady drop (rise) in its performance", and that "Profits, sales, and orders have dropped (risen). Experts hold bad (strong) management responsible for this drop (rise)." This information was accompanied by a graph showing a drop (increase) in sales. In the two poor company performance conditions, another paragraph was displayed describing the type of crisis. Participants either read about a *relational crisis* ("this crisis was mainly relational due to inappropriate people management. Poor handling of internal disharmony in a competitive market environment had seriously damaged employees' relations and motivations.") or a *financial crisis* ("this crisis was mainly financial due to inappropriate financial management. Poor financial choices in a competitive market environment had seriously damaged the company's marketplace position.").

After reading the organizational scenario, participants answered manipulation check questions and evaluated the importance of leadership behaviors and traits for the future CEO of this company. Then participants were asked to evaluate a list of job candidates (see Appendix A3 Table in S1 Appendix) and to select the most suitable candidate to ensure appropriate organizational functioning of the company: "The company has made a preselection of candidates for this high-profile position. Below you will find descriptions of four qualified candidates. These descriptions are the result of interviews and psychological tests. Please carefully read each description. You will then be asked to evaluate these candidates and choose the one who is best suited to ensure appropriate functioning of the company." Participants were thus asked to take the perspective of the company and make the best decision for this company. For each candidate, participants were then asked to evaluate whether they would be a suitable leader and choose one from a list of four job candidates who differed by gender (male versus female) and in terms of gendered traits (agentic versus communal), using the same brief CVs as in Study 1. Candidate gender was manipulated with typically male or female first names taken from research using English names [43]. The combination of the two trait descriptions with the two genders of the candidates was counterbalanced so that each trait description (agentic versus communal) appeared with the female and the male candidates for half of the participants. Finally, in Study 2, participants had to rate their selected candidate according to their potential to signal or effect actual change [37, 43].

**Measures.** In the following section, all measures are described in chronology of appearance. Response scales for leadership behaviors and traits ranged from 1 (*Not at all important)* to 7 (*Very important)*, for other items from 1 (*Strongly disagree)* to 7 (*Strongly agree)*. Additional measures are presented in the S1 File (leader suitability, relevance of agentic behavior and traits, actual change).

*Manipulation checks.* After reading the scenarios, participants were asked to respond on a 7-point scale whether "The performance of this company is . . ." 1 = *good* to 7 = *bad*. Two items also checked whether participants correctly retained the task of the new CEO in the relational versus financial crisis situations: "The main aim of hiring a new CEO is to find the right

person who knows how to improve the financial performance of the company" and "The main aim of hiring a new CEO is to find the right person who knows how to improve the relations between employees in the company".

*Communal leadership attributes.* The same scales as is Study 1 were presented: the Competing Values Management Practices Survey [59] measured four communal behaviors ($\alpha$ = .90; $M$ = 5.17, $SD$ = 1.37) and the PAQ scale [60] measured eight communal traits ($\alpha$ = .90; $M$ = 5.23, $SD$ = 1.04).

*Signaling change.* After candidate selection, several items measuring change potential were presented (see all items in the S5 Table in S1 File). Four items measured the degree to which the appointment was made to signal change ($\alpha$ = .90; $M$ = 5.60, $SD$ = 1.23; "The fact of appointing this candidate will show that the company wants to change the type of management. The choice of this candidate symbolizes a visible change for partners and competitors." from [37]; and "The choice of this candidate signals to investors that Jefferson is willing to substantially change things. Choosing this candidate as CEO for Jefferson symbolizes the start of a new era." from [43]).

## Results

**Preliminary analyses.**   For the three manipulation check items, we performed three ANOVAs with crisis type (relational versus financial versus no crisis) as between-subject factor. The ANOVAs yielded significant variations between crisis types (performance check: $F$ (2,381) = 466.22, $p$ < .001, $\eta_p^2$ = .71; relational check: $F$ (2,381) = 47.74, $p$ < .001, $\eta_p^2$ = .20; financial check: $F$ (2,381) = 3.06, $p$ = .048, $\eta_p^2$ = .02). Post-hoc tests with Bonferroni correction showed that performance was perceived as better in the no crisis ($M$ = 1.92; $SD$ = 1.38) compared to both the relational ($M$ = 6.24, $SD$ = 1.09, $p$ < .001, 95% CI [-4.71, -3.94]) and the financial crisis conditions ($M$ = 6.03, $SD$ = 1.32, $p$ < .001, 95% CI [-4.50, -3.73]), that the task of the new CEO was more strongly expected to improve relations in the relational crisis ($M$ = 6.03; $SD$ = 1.34) than in both the financial ($M$ = 4.06, $SD$ = 1.90, $p$ < .001, 95% CI [1.47, 2.44]) and the no crisis conditions ($M$ = 4.85, $SD$ = 1.60, $p$ < .001, 95% CI [0.69, 1.67]), and that the new CEO was more strongly expected to improve the financial performance in the financial crisis ($M$ = 5.80; $SD$ = 1.66) than in the relational condition ($M$ = 5.31, $SD$ = 1.72, $p$ = .050, 95% CI [0.000, 0.96]), but not compared to the no crisis condition ($M$ = 5.46; $SD$ = 1.42, $p$ = .276, 95% CI [-0.15, 0.83]). Our manipulations were therefore successful, with the addendum that participants assumed that a CEO should improve financial performance whether the company was in a financial crisis or not.

Participant gender had no significant main or interaction effect on any of the analyses presented below, we thus report analyses without participant gender.

We first looked at general preferences for the four candidates. The agentic female candidate was chosen by 38%, the agentic male by 25.8%, the communal female by 20.1% and the communal male by 16.1%.

**Preferences for candidate gendered traits.**   H1 predicted that the preference for communal leaders in crisis situations over non-crisis contexts would only occur in a relational crisis scenario. We conducted a logistic regression on candidate gendered traits (0 = choice of an agentic candidate, 1 = communal candidate). Following our prediction, we entered two orthogonal contrasts for crisis type (Contrast 1: 1 = relational crisis, 0 = no crisis, -1 = financial crisis; residual Contrast 2: 1 = relational and financial crises, -2 = no crisis). In support of H1, a C1 crisis type effect showed that overall participants were more likely to select communal leaders when exposed to the relational crisis (55.3%) than to the financial crisis (20.9%) conditions, $B$ = 0.77, $\chi^2$ (1, N = 384) = 30.69, $p$ < .001, $e^B$ = 2.16, 95% CI [1.65, 2.84], with the no crisis

(31.7%) situated in-between as the residual C2 effect was not significant, $B = 0.07$, $\chi^2$ (1, N = 384) = 0.77, $p = .381$, $e^B = 1.07$, 95% CI [0.92, 1.25] (Table 2). Overall, the agentic candidates (63.8%) were more likely to be selected than communal candidates (36.2%), $B = -0.63$, $\chi^2$ (1, N = 384) = 30.84, $p < .001$, $e^B = 0.53$.

**Preferences for candidate gender.** We conducted a logistic regression on candidate gender (0 = choice of a male candidate, 1 = female candidate). Following our prediction in H2 where choices in the relational crisis would be higher than in the control condition, with the financial crisis situated in-between, we entered two orthogonal contrasts for crisis type (Contrast 1: 1 = relational crisis, 0 = financial crisis, -1 = no crisis; residual Contrast 2: 1 = relational crisis and no crisis, -2 = financial crisis). The results for H2 regarding the expected main effect of crisis type C1 were in the expected direction (relational crisis 65.2%; financial crisis 55.8%; no-crisis: 52.8%), $B = 0.26$, $\chi^2$ (1, N = 384) = 3.97, $p = .046$, $e^B = 1.30$, 95% CI [1.00, 1.66] (Table 2). The residual C2 effect was not significant, $B = 0.05$, $\chi^2$ (1, N = 384) = 0.39, $p = .534$ $e^B = 1.05$, 95% CI [0.91, 1.21]. Overall, female candidates (58.1%) were more likely to be selected than male candidates (42.9%), $B = 0.32$, $\chi^2$ (1, N = 384) = 9.72, $p = .002$, $e^B = 1.38$.

**Perceived relevance of leadership attributes.** For a test of H3, we first performed two separate ANOVAs, with both communal leadership behavior and traits as outcome variables. The C1 contrast (1 = relational crisis, 0 = no crisis, -1 = financial crisis) was entered as the independent variable, while controlling for C2 (1 = relational crisis and financial crisis, -2 = no crisis).

For communal leadership behavior, a crisis type C1 effect was shown, $F$ (1, 381) = 22.69, $p < .001$, $\eta_p^2 = .06$, indicating that in the relational crisis ($M = 5.60$, $SE = 0.12$) communal behavior was perceived as more relevant than in the financial crisis ($M = 4.81$, $SE = 0.12$), with no crisis ($M = 5.09$, $SE = 0.12$) situated in-between (i.e., the C2 effect was not significant, $F$ (1, 381) = 0.56, $p = .454$, $\eta_p^2 = .001$).

Similarly, for communal leadership traits, a crisis type C1 effect was shown, $F$ (1, 381) = 11.24, $p < .001$, $\eta_p^2 = .03$, indicating that in the relational crisis ($M = 5.48$, $SE = 0.09$) communal traits were perceived as more relevant than in the financial crisis ($M = 5.05$, $SE = 0.09$), with no crisis ($M = 5.15$, $SE = 0.09$) situated in-between (i.e., the C2 effect was not significant, $F$ (1, 381) = 1.17, $p = .279$, $\eta_p^2 = .003$).

**Mediational analyses.** In order to fully test whether the relevance of communal attributes could explain the preferential choice for a communal leader in a relational crisis, as compared to a financial crisis, with the no crisis condition situated in-between (H3), we ran mediation model 4 using Hayes' [61] PROCESS macro with 10,000 biased bootstrap samples. PROCESS can estimate models with binary outcomes and estimate accordingly. The dependent measure was candidate gendered traits (0 = agentic, 1 = communal). The C1 contrast of crisis type (1 = relational crisis, 0 = no crisis, -1 = financial crisis) was entered as the independent variable, while controlling for C2 (1 = relational crisis and financial crisis, -2 = no crisis). Communal leadership behavior and traits were posed as simultaneous mediators (see Fig 2).

**Table 2. Choice of a communal versus agentic and female versus male CEO as a function of type of crisis (Study 2, United States of America).**

|  | Choice of candidates % (n) | | | |
| --- | --- | --- | --- | --- |
|  | Relational crisis | Financial crisis | No crisis | Total |
| Communal candidates | 55.3 (73) | 20.9 (27) | 31.7 (39) | 36.2 (139) |
| Agentic candidates | 44.7 (59) | 79.1 (102) | 68.3 (84) | 63.8 (245) |
| Total | 100.0 (132) | 100.0 (129) | 100.0 (123) | 100.0 (384) |
| Female candidates | 65.2 (86) | 55.8 (72) | 52.8 (65) | 58.1 (223) |
| Male candidates | 34.8 (46) | 44.2 (57) | 47.2 (58) | 41.9 (161) |
| Total | 100.0 (132) | 100.0 (129) | 100.0 (123) | 100.0 (384) |

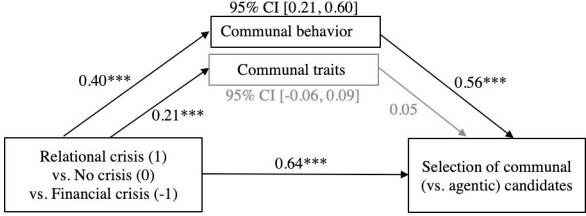

**Fig 2. Indirect effect of crisis type (contrast C1), mediated by communal leadership behavior but not communal leadership traits, on the preference for communal (as opposed to agentic) candidates (Study 2, United States of America).** Process Model 4 of Hayes (2018, [62]). Numbers indicate unstandardized coefficients. *** p < .001.

For leadership behavior, path 'a' showed that the incremental effect of financial crisis—no crisis—relational crisis (C1) was related to higher relevance ratings of communal behavior, $B = 0.40$, $SE = 0.08$, $p < .001$, 95% CI [0.23, 0.56], and communal traits, $B = 0.21$, $SE = 0.06$, $p < .001$, 95% CI [0.09, 0.34]. Path 'b' showed that the more communal behavior was rated as relevant, the more likely a communal candidate would be to be chosen, $B = 0.95$, $SE = .14$, $p < .001$, 95% CI [0.67, 1.21]. Conversely, path 'b' for communal traits was not significant, $B = 0.06$, $SE = .08$, $p = .72$, 95% CI [-0.25, 0.63]. The direct effect of crisis type C1 on candidate selection was significant, path 'c': $B = 0.64$, $SE = 0.15$, $p < .001$, 95% CI [0.35, 0.93], as was the indirect effect through communal behavior, $B = 0.22$, $SE = 0.07$, 95% CI [0.11, 0.38]. The indirect effect through communal traits was not significant, $B = 0.01$, $SE = 0.40$, 95% CI [-0.06, 0.09].

In line with Hypothesis 3, ratings of relevance for communal behavior explained the stronger preference of communal candidates in a relational crisis versus no crisis versus financial crisis. In contrast, communal traits had no mediating effect, in only partial support of H3.

**Additional analyses: Change potential.** Previous research suggests that in times of crisis the atypicality of a candidate may be valued and used to signal change [37]. In order to explore whether the potential to signaling change was associated more with candidate choice in crisis compared to no crisis contexts, we performed an ANOVA on signaling change with orthogonal contrasts for crisis type as predictors. C1 opposes the two crisis conditions to the no crisis (1 = relational and financial crisis, -2 no crisis) and thus tests our exploratory question, and C2 tests for differences between the two crisis conditions (1 = relational, -1 = financial crisis, 0 = no crisis). As expected crisis type C1 had an effect, $B = 0.36$, $SE = 0.04$, $t(381) = 8.67$, $p < .001$, 95% CI [0.28, 0.44], $\eta_p^2 = .17$, revealing that signaling change potential was more strongly associated with candidates chosen in the relational ($M = 5.90$, $SE = 0.10$) and financial crises ($M = 5.99$, $SE = 0.10$) than in the no crisis condition ($M = 4.88$, $SE = 0.10$). C2 was not significant, thus the two crisis conditions did not differ, $B = -0.05$, $SE = 0.07$, $t(381) = -0.64$, $p = .521$, 95% CI [-0.18, 0.09], $\eta_p^2 = .001$.

We performed PROCESS model 4 mediational analysis [61], testing the role of signaling change as a mediator for the choice of candidate gender depending on crisis type. We opted for a contrast-coding that opposed the two crisis types to the no crisis (C1), controlling for the residual orthogonal contrast (C2). The mediation effect was not significant, $B = -0.01$, $SE = 0.03$, 95% CI [-0.08, 0.06]. Thus, although signaling change was associated more with candidates chosen in both crisis contexts, path 'a': $B = 0.36$, $SE = 0.04$, $p < .001$, 95% CI [0.27, 0.44], it was not specifically more associated with the choice of a woman, path 'b': $B = -0.03$, $SE = 0.09$, $p = .775$, 95% CI [-0.21, 0.16]. We acknowledge that the causality path is ambiguous because signal of change was measured after candidate choice. Because of this, ratings of signaling change could also be due to post decision justification. In Study 3, we thus changed the study design, allowing us to measure a candidate's potential to signaling change before candidate choice, in order to have greater confidence in the causality of mediation paths.

## Discussion

This simplified study replicated the patterns found in Study 1 with a larger sample size, by showing that both communal candidates and female candidates had a leadership advantage in the relational crisis compared to the other two conditions. As expected, while the financial crisis was the least preferred condition for communal candidates (consistent with H1), the no crisis condition was the least preferred for female candidates (consistent with H2). Moreover, Studies 1 and 2, as well as [43]—showed that the choice of gendered traits was more strongly affected by organizational context than candidate gender, suggesting that participants are more likely to focus more on leadership style characteristics than stereotypes associated with gender.

The distinct patterns for gendered traits and gender point to different mechanisms at play. Indeed, communal attributes, in terms of leadership behavior (Studies 1 and 2), were most strongly demanded in a relational crisis, and least in a financial crisis, thereby explaining the choice of communal candidates, but they did not explain the choice of a woman. This comes without surprise as traits are considered stable personality entities and are more likely to be associated with candidate gender. Hence the relevance of communal behavior, which can be learned and flexibly adapted, may be stronger in predicting a choice for communal candidates, who may be perceived to have acquired these traits throughout experience (irrespective of perceptions of their inherited traits in link with their biological sex). In contrast, because female leaders may be perceived as having inherited unchangeable traits–such as communal traits– these stable traits may not guide participant choice, but rather are overshadowed by other factors such as the potential to signaling change.

Our additional analyses revealed that the potential to signaling change was more strongly associated with candidates chosen in the crisis conditions than those chosen in the no crisis condition. However, mediational analyses did not show evidence that particularly the choice of female leaders was affected by the signaling change motive. The motivation to signal change likely has some impact on candidate choice, but other factors may outweigh it when it comes to choice based on gender.

## Study 3

This study was run for two purposes: First, we intended to replicate the findings from the first two studies by inverting the experimental procedure, that is, present one out of the four candidates to each participant and ask them to choose from the three organizational contexts (relational, financial, and no crisis). Second, we wanted to explore the role of potential for change in a better suited experimental design, allowing us to measure the signal of change potential associated with the candidate before choosing the organizational context. If significant, the causality of a candidate's potential for change as predictive of the choice of crisis context could be assumed.

## Method

Ethic approval was obtained from the first author's faculty ethic comity for Study 3 (Faculty of Psychology and Educational Science of the University of Geneva, modification of N˚ PSE.20190305.06). Participants gave written informed consent by clicking on one of two possible answers (Yes or No) at the beginning of the study just after reading general information about study content and data management, and again at the end of the questionnaire after having been fully debriefed about the experimental manipulations and aims of the study.

**Participants.** Based on the same arguments discussed in Study 2, we considered again a small effect size for the *a priori* power analysis. Thus, in order to have 80% power to capture a

minimum effect size of an Odd Ratio = 1.80 (with a probability under H0, p1 = .50), we aimed for a final sample size of 375 participants. We requested for 450 participants and received $N$ = 447 completed questionnaires. For this study, the entry criteria were more rigorous, as we asked participants to have at least three years of work experience while being employed by a company (Study 2 had included participants who were self-employed, retired, or in education). Participants were $N$ = 385 MTurk workers after exclusion of 62 participants. A sensitivity analysis for logistic regressions revealed that with $N$ = 385, $\alpha$ = .05, and 80% desired power, the minimum effect size that we could detect was an Odd Ratio = 1.78 (based on a probability under H0, p1 = 0.5). Exclusion criteria were the following: Participants who did not give informed consent after the debriefing at the end ($n$ = 17), who did not pass the attention check ($n$ = 10), who got one or several of the manipulation checks wrong ($n$ = 34; $n$ = 11 participants answered incorrectly the item about which of the companies "has the best performance", $n$ = 17 answered incorrectly to the item asking which of the companies, "is going through problems due to its financial performance", and $n$ = 14 answered incorrectly to the item asking which of the companies, "is going through problems due to relationships between employees in the company"), who did not indicate their gender ($n$ = 3), and who were not English native speakers ($n$ = 11).

Participants included 57.1% women and reflected ethnic composition of the USA (68.8% White, 15.3% Hispanic/Latino; 7% Asian American; 10.1% African-American; 1% Native American, 0.5% other). Participants were on average $M$ = 38.20 years old ($SD$ = 11.59; minimum 19, maximum 72 years), and 82.3% had at least an Associate degree. Participants had on average $M$ = 6.78 years of work-experience with their current employer ($SD$ = 6.16, minimum 0 year, maximum 35 years), and $M$ = 17.99 years of work-experience overall ($SD$ = 11.33, minimum 3 year, maximum 54 years). All participants were employed; 88.6% worked full-time, and 58.4% held a management position.

**Procedure.** The study had a 2 (Candidate gendered traits: agentic, communal) × 2 (Candidate gender: male, female) between participant design with choice of crisis type (relational crisis, financial crisis, no crisis) as the outcome variable.

After consenting to participate, participants read three fictitious articles about the companies "Campbell", "Jefferson", and "Morton" who are each seeking a new CEO (see Appendix A2a, A2b, and A2c Fig in S1 Appendix). We used the same materials as in Study 2 (apart from small wording modifications so the three texts were not identical) in order to introduce the three organizational contexts. The order and the naming of the companies were randomized. Participants then responded to manipulation check items and rated which leadership behaviors were relevant for each one of the companies. Finally, participants were presented with one out of the four candidates from Study 2, again counter-balancing the two communal and the two agentic profiles for the male and female candidates (see Appendix A4 Table in S1 Appendix). Participants assessed the candidate's signaling change potential and then received the following instructions: "The headhunter can recommend [Alan/Claire] Jones only for one of the three companies. The candidate should ensure an efficient functioning and a strong performance of the company. If you were to advise the headhunter, which company would you recommend Jones for:". Then participants could indicate their choice between the three companies which appeared with their names and the articles and justify their choice with the leader suitability items (which are reported in the S1 File).

**Measures.** *Manipulation checks.* After reading the three scenarios, participants were asked which of the companies ". . .has the best performance", ". . . is going through problems due to its financial performance?", and ". . .is going through problems due to relationships between employees in the company?" with three answer options (Campbell, Jefferson, or Morton).

*Communal leadership behaviors*. On a new page, participants saw each article again and had to rate on the same scale as in the two previous studies the relevant leadership behaviors for each company ([59]; communal behavior, for all companies α > .90; relational crisis: *M* = 5.96, *SD* = 1.39, financial crisis: *M* = 4.99, *SD* = 1.31, no crisis *M* = 5.47, *SD* = 1.18).

*Signaling change*. As in Study 2, the four signaling change items were presented (For all candidates αs > .88; male agentic: *M* = 5.40, *SD* = 1.12, female agentic: *M* = 5.60, *SD* = 1.12, male communal: *M* = 5.59, *SD* = 1.00, female communal: *M* = 5.49, *SD* = 1.01).

## Results

**Preliminary analyses.** We controlled for participant gender as a main effect and in all interactions in all analyses presented here. It had no significant effect in any analysis, we thus did not consider participant gender in the analyses presented hereafter.

We first looked at general preferences for the three companies. The relational crisis was chosen by 56.6%, the financial crisis by 29.1%, and the no crisis context by 14.3%.

**Preferences for crisis type.** We conducted two multinominal logistic regressions entering candidate gendered traits (-1 = agentic, 1 = communal candidates), candidate gender (-1 = male, 1 = female candidates), and their interaction as predictors, and crisis type as the outcome variable. In the first multinomial logistic regression, we specified the no crisis condition as the reference category. Thus, in this logistic regression model we were interested in the probabilities of choosing the relational crisis (versus no crisis), and, respectively, the financial crisis (versus no crisis), as a function of the predictors. In the second multinomial logistic regression, we specified the financial crisis condition as the reference category. This regression model informed us about the probability of choosing the relational crisis (versus financial crisis) and, respectively, the no crisis (versus financial crisis) situation, as a function of the predictors.

Regarding candidate gendered traits, the results fully support H1. The probability of choosing the relational crisis (versus no crisis) was higher for the communal candidate compared to the agentic candidate, $B = 1.04$, $\chi^2$ (1, N = 385) = 34.88, $p < .001$, $e^B = 2.84$, 95% CI [2.01, 4.01]. Furthermore, the probability of choosing the no crisis situation (versus financial crisis) was higher for the communal candidate compared to the agentic candidate, $B = 1.03$, $\chi^2$ (1, N = 385) = 15.07, $p < .001$, $e^B = 2.80$, 95% CI [1.66, 4.70], and, the choice of the relational crisis (versus the financial crisis) was higher for the communal candidate compared to the agentic candidate, $B = 2.07$, $\chi^2$ (1, N = 385) = 80.66, $p < .001$, $e^B = 7.94$, 95% CI [5.05, 12.47]. Table 3 presents the percentages of choice for each type of crisis as a function of candidate gendered traits.

Regarding candidate gender, the first regression showed that the probability of choosing the relational crisis context (versus no crisis) was higher for the female candidate compared to

**Table 3. Choice of a relational crisis versus financial crisis versus no crisis as a function of candidate gendered traits and candidate gender (Study 3, United States of America).**

|  | Choice of crisis type % (n) | | | |
|---|---|---|---|---|
|  | Relational crisis | Financial crisis | No crisis | Total |
| Communal candidates | 86.9 (172) | 3.5 (7) | 9.6 (19) | 100.0 (198) |
| Agentic candidates | 24.6 (46) | 56.1 (105) | 19.3 (36) | 100.0 (187) |
| Total | 56.6 (218) | 29.1 (112) | 14.3 (55) | 100.0 (385) |
| Female candidates | 61.3 (117) | 28.3 (54) | 10.5 (20) | 100.0 (191) |
| Male candidates | 52.1 (101) | 29.9 (58) | 18.0 (35) | 100.0 (194) |
| Total | 56.6 (218) | 29.1 (112) | 14.3 (55) | 100.0 (385) |

the male candidate, $B = 0.45$, $\chi^2$ (1, N = 385) = 6.57, $p = .010$, $e^B = 1.57$, 95% CI [1.11, 2.22]. However, the probability of choosing the relational crisis (compared to the financial crisis) did not differ according to candidate gender, $B = 0.35$, $\chi^2$ (1, N = 385) = 2.24, $p = .135$, $e^B = 1.41$, 95% CI [0.90, 2.22]. Finally, the probability of choosing the financial context (versus no crisis) did not differ for the female candidate compared to the male candidate, $B = 0.11$, $\chi^2$ (1, N = 385) = 0.17, $p = .684$, $e^B = 1.11$, 95% CI [0.66, 1.87]. Table 3 presents the percentages of choice for each type of crisis as a function of candidate gender. Thus, only partial support was found for the effect predicted in H2, with a significant difference regarding the probability of choosing the relational crisis versus no crisis contexts.

The candidate gendered traits and gender interaction was not significant in any of the analyses (all $p$s > .489).

**Perceived relevance of leadership attributes.** The design of Study 3 did not allow us to test Hypothesis 3, in terms of a classical mediation where the relation between candidate gendered traits and the type of crisis selected is mediated by the perceived relevance of leadership behaviors. We therefore conducted two complementary sets of analyses that offer support for our theoretical reasoning. We first investigated whether the communal leadership behavior was stronger for participants who had selected the relational crisis compared to those who had selected the no crisis, and compared to those who had selected the financial crisis. We performed a within-participant ANOVA with relevance ratings of communal leadership behavior for the three crisis types as repeated measures. Communal leadership behavior was deemed more important in the relational crisis ($M = 5.96$, $SE = 0.07$) compared to a no crisis situation ($M = 5.47$, $SE = 0.06$), $F$ (1, 384) = 39.60, $p < .001$, $\eta_p^2 = .09$, which was different from the financial crisis ($M = 4.99$, $SE = 0.07$), $F$ (1, 384) = 69.20, $p < .001$, $\eta_p^2 = .15$. Second, we investigated whether a candidate's gendered traits were linked to the perceived relevance of the leadership behavior for the type of crisis matched to the candidate. We conducted an ANOVA with candidate gendered traits and candidate gender as between-participant factors. The dependent variable was the communal behavior for the organizational context preferred for the candidate. Supporting the reasoning of H3, the main effect of candidate gendered traits was significant, $B = 0.29$, $SE = 0.07$, $t(381) = 4.12$, $p < .001$, 95% CI [0.15, 0.43], $\eta_p^2 = .04$. The perceived relevance of the communal behavior was higher for the type of crisis matched to the communal candidate ($M = 5.82$, $SE = 0.10$) compared to the one matched to the agentic candidate ($M = 5.24$, $SE = 0.10$). No other effects were significant, $p$s > .194.

**Additional analyses: Signaling change.** In order to explore the effects of the potential to signaling change, we performed an ANOVA with candidate gendered traits and candidate gender as between-participant factors. No effect was significant, all $p$s > .171.

## Discussion

In the third study, we applied an inverted experimental design to check the robustness of our findings and to further explore the role of signaling change. Studies 1 and 2 presented the four candidate types to participants, while only showing one organizational condition, whereas the final study revealed all three organizational contexts but not the different candidate profiles. These variations allowed us to test the robustness of our findings. Consistent with the first two studies and H1, a strong effect showed that the communal candidate—relational crisis association was stronger than the communal—no crisis association, with the communal—financial crisis association being the weakest. For candidate gender, the effects found also followed patterns from Studies 1 and 2. The female-relational crisis preference was stronger compared to the no crisis condition, with the financial crisis situated in-between but not significantly different from either. Thus, this progressive pattern was in the predicted direction; however, results

were weak, thus only partially supporting H2. This was in line with prior research [43] illustrating that in experimental settings, candidate gender plays a weaker role in decision making in the presence of information about the gendered trait characteristics of a candidate.

We find no support for signaling change as more strongly associated with women, nor evidence for its role as a motivator underlying a female-crisis association. Participants were asked to evaluate candidates according to their potential to signaling change and only afterwards they chose a suitable company context. Following this chronology, participants might not have viewed the potential to signal change in the context of the leadership mission expected from the candidate. It may therefore be a better strategy to ask participants to match the candidate with a company, and then to outline their reasoning for their choice.

## General discussion

Glass cliff research has shown that women are preferred in organizations suffering from a crisis [8–10]. Moreover, previous findings suggest that in crisis situations the traditional think manager-think male stereotype may be less consistent [13]. In the present work, we provided empirical evidence establishing that a glass cliff preference for women, and for leaders ascribed stereotypically feminine traits, occurs particularly in specific types of crises which are relational in nature. We examined variations in how people's stereotypes about leadership effectiveness vary across different types of crisis, and identified the perceptual contingencies governing crises due to "feminine" (e.g., relational) versus "masculine" (e.g., financial) problems. Results from three experiments consistently showed that the choice of female gendered leaders and leaders with feminine gendered (communal) traits are not driven by the same motivations. Whereas communal traits were strongly associated with a relational crisis, and higher ratings of the relevance of communal leadership behavior drove this effect, choices for the female gender were more complex. Although, like feminine traits, female gender was also more strongly associated with a relational than a no crisis context, the financial crisis was situated in-between the relational and no crisis situations. These findings suggest that for the choice of women, their stereotypically associated feminine (or communal) traits may be deemed useful in a crisis demanding relational competences, but other factors also play a role. One such factor, the choice of women for their atypicality or potential to signaling change was explored, but evidence was mitigated.

By investigating variations in crisis typology, our findings add to explanations for the glass cliff. Specifically, beyond extending past research that has identified motivating factors for the glass cliff which do not value women's or communal competences [13, 63], we investigated crisis contexts where stereotypically feminine qualities are perceived as an added value for leadership effectiveness. Although the mechanism explaining the think crisis-think *feminine* association has been understood in the present research, the underpinnings of the think crisis-think *female* association are more complex. It seems that other factors which concern the female gender particularly, such as their atypicality and signaling change purpose [37], might shape part of the motivation for the selection of women. We tested this idea, with Study 2 showing an association of a motivation to signaling change with the choice of candidates in crisis contexts. However, this was true for both the choice of female and male candidates. In Study 3, we made a further attempt to establish a link to a signaling change explanation by asking participants to first rate candidates on their potential to signaling change, and then to attribute them to a company context. No conclusive results were revealed, probably due to the lack of meaning of the signaling idea if a candidate is not evaluated on the background of a specific (crisis-) mission. Future research should investigate whether what is understood by signaling change is understood in different ways for men and women. Moreover, a direct test of the

impact of signaling change intentions could be performed by manipulating or priming participants with a mission to signaling change, with a consecutive measure of their leadership preferences in terms of gender. The analysis of audience reactions and their anticipation by decision-makers is an important field for future research. For example, recent research has suggested that audience reactions to organizational failure differed in relation to leader gender and failure type [64].

We found that when a leader's tasks involved dealing with financial problems, or competition, agentic leaders were the most selected. These findings extend research suggesting that stereotypically male qualities are deemed useful in many crisis contexts [43], and are also largely consistent with experimental work showing that communal orientations elicit negative evaluations of leadership effectiveness when organizational tasks involve economic and financial functions [42].

It is of interest to note that despite a stronger preference for female CEOs in crisis than no crisis contexts, overall male CEOs were preferred in Study 1 in a Spain sample. In Study 2 with a US sample, no main effect of gender was observed, suggesting that in this context, female leadership has potentially become a more accessible, or acceptable concept. In the corporate world, relatively few women are found in top leadership positions, and particularly in Spain, management is still perceived as less compatible with female gender compared to other European countries and the United States of America [65].

An interesting observation provided by the last study is that gender and gendered traits did not interact in the female-crisis and feminine-crisis associations. However, future research should also focus on more androgynous profiles of leaders, that is, leaders having communal and agentic traits. The modern leadership literature suggests that the ideal manager is one who can pivot between both agentic and communal behavioral and trait dimensions, and that women may have a particular advantage when it comes to displaying androgynous trait combinations (e.g., [66]). However, this literature has investigated these associations without considering organizational features such as performance [67]. The present experiments were deliberately limited to opposing gendered trait variants (agency OR communion) in order to establish, as a first step, the pure unadulterated contributions of the two gender and gendered trait dimensions. However, the reality is much more fluid, and future work should focus on more complex variations of gender and gendered trait dimensions. Moreover, intersectionality with other social categories such as ethnicity in the context of the glass cliff may bear interesting insights, see for example [39].

## Implications

The present work on the association of leader characteristics in terms of gender, and gendered traits in the context of organizational performance, has at least two important practical implications. First, there is a direct application to an increasing number of organizations that are unexpectedly engaged in different types of crisis situations, which often involve dealing with financial and economic goals. Given that people's leadership schemas influence their perceptions of leaders as well as relevant outcomes, such as job satisfaction, organizational commitment or well-being [68], our findings serve to make more nuanced sense of managerial behavior in difficult organizational settings. Second, our findings have important implications for organizational performance in crisis management. As the present findings suggest, communal qualities–and to a smaller extent women—are perceived as a sign of lower performance in organizational situations that explicitly involve economic problems. However, previous studies have demonstrated that stereotypically feminine leadership abilities, such as promoting teamwork, joint effort, and shared goals are key elements in transforming industries in crisis,

not only when the problem is internal disharmony but also when broader financial factors are involved [19, 20, 69]. Stereotypical beliefs that leaders with stereotypically masculine traits are more appropriate for leading in such crisis contexts thus run the risk of reinforcing less collaborative and more hierarchical leadership styles during times when interpersonally oriented, "feminine" leadership characteristics are indeed particularly relevant.

## Conclusion

Our findings establish that women and candidates with communal qualities are more likely to access leadership positions in crisis contexts that call for communal qualities, than in financial crisis or no crisis contexts. However, more studies are needed to disentangle the complexities of stereotypes and decision-making regarding selection of women versus candidates with communal qualities in crisis contexts, as well as their effects for career development. A key implication of the current study is that the stereotypes that define effective leadership in crisis management are not necessarily linked to women and communal dimensions, as one would expect from glass cliff effects. When an organization facing a relational crisis is seeking a competent leader, a think manager-think feminine stereotype seems to apply. However, when an organization is choosing a female leader, communal competences are not the only determining factors.

## Supporting information

**S1 Appendix. Appendix.**
(DOCX)

**S1 File. Supporting information.**
(DOCX)

## Acknowledgments

We thank Sarah Robinson for her proof-reading comments on the final version of this paper.

## Author Contributions

**Conceptualization:** Leire Gartzia, Meera Komarraju.

**Data curation:** Clara Kulich, Leire Gartzia.

**Formal analysis:** Clara Kulich, Cristina Aelenei.

**Funding acquisition:** Clara Kulich.

**Investigation:** Clara Kulich.

**Methodology:** Clara Kulich, Cristina Aelenei.

**Project administration:** Clara Kulich.

**Writing – original draft:** Clara Kulich, Leire Gartzia.

**Writing – review & editing:** Clara Kulich, Leire Gartzia, Meera Komarraju, Cristina Aelenei.

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
