## [Decision Letter · Decision Letter 0]

30 Jul 2020

PONE-D-20-16233

Contextualizing the Think Crisis-Think Female Stereotype in Explaining the Glass Cliff Gendered Traits, Gender, and Type of Crisis

PLOS ONE

Dear Dr. Kulich,

Thank you for submitting your manuscript to PLOS ONE. After careful consideration, we feel that it has merit but does not fully meet PLOS ONE’s publication criteria as it currently stands. Therefore, we invite you to submit a revised version of the manuscript that addresses the points raised during the review process.

We look forward to receiving your revised manuscript.

Kind regards,

I-Ching Lee

Academic Editor

PLOS ONE

Journal Requirements:

2. Please include the full name of the Institutional Review Board that approved your study in the ethics statement.

Additional Editor Comments (if provided):

I have the reviews back from experts in the field and they generally see the merits of the manuscript. However, both reviewers also raise major concerns over the current manuscript, which I completely agree. I'll raise these major concerns in the following but please also pay attention to minor concerns each individual reviewer raised.

1. Problems with the data and analyses

Data: In Study 1, you excluded participants who chose the extreme agentic filler candidate. I disagree with this choice (also see comment from Reviewer 1). You cannot exclude participants just because they do not respond the way you expect them to unless there are sufficient reasons for doing so (e.g., those who do not pass the manipulation check). Thus, I would like you to put these participants back to your data set.

Data analysis: In Study 1, you include two contrasts in the regression models. However, it is not clear to me how the Contrast 2 could be interpreted (also see comment from Reviewer 1). I assume that the inclusion of the Contrast 2 is to ascertain whether the two experimental conditions differ from the control condition but not in the way you expected (which was tested in Contrast 1). In this case, the contrast should be 1 = relational crisis, 0 = no crisis, 1 = financial crisis so that the intercept is more interpretable. In addition, the interaction term (gender traits x candidate sex) was not tested consistently across studies (see comment from Reviewer 2). Please test it across all studies. In addition, if the contrast 1 is significant, please indicate whether the two comparisons were both significant (relational crisis vs. no crisis; financial crisis vs. no crisis) or not. Lastly, I agree with the Reviewer 1 that you should be cautious about using difference scores. It would be better to test them separately because communal and agency traits are of two dimensions rather than two poles of one dimension.

2. Problems with presentation

c. literature gaps: There are some choices made in the method section in which readers have not been informed (e.g., the organizational roles; the distinction of traits and behaviors).

b. Please reorder the information, move the sensitivity test to the method section under participants (because the information pertains to the sample size) and address the concern raised by Reviewer 1.

c. I would rather see tables with all the results than the bar figures that contain very little information. Right now, we only get a glimpse of the logistic models, without seeing the whole picture.

d. please list the case number for the ethical approval in the text.

e. Whenever a main effect appears, please conduct post hoc comparisons to ascertain whether the two experimental conditions differ from the control condition (e.g., on page 30, only mean scores were reported without the p-values for the comparisons).

f. I am very confused in terms of how the analysis was done done in Study 3. You stated that “We conducted multinominal logistic regressions…” on line 870 but listed “crisis type as outcome variable (1 = relational crisis, 2 = financial crisis, 3 = no crisis…” on lines 872-875. You further reported findings in the first regression model and stated that “the probability to choose the relational crisis (versus no crisis) on lines 876 – 877 and further “the probability to choose the no crisis (versus final crisis)… on line 879. I am not sure how categorical variables can be numbered and entered in a regression model and why the reference group kept changing in the findings of the same regression model. Findings in Study 3 should be reported so readers understand what was done and can interpret the findings themselves.

g. There are places that it’s quite difficult to follow/read. Please carefully read your manuscript and revise accordingly or seek editing services.

Reviewers' comments:

Reviewer's Responses to Questions

**Comments to the Author**

1. Is the manuscript technically sound, and do the data support the conclusions?

Reviewer #1: Partly

Reviewer #2: Partly

2. Has the statistical analysis been performed appropriately and rigorously? 

Reviewer #1: I Don't Know

Reviewer #2: I Don't Know

3. Have the authors made all data underlying the findings in their manuscript fully available?

Reviewer #1: Yes

Reviewer #2: Yes

4. Is the manuscript presented in an intelligible fashion and written in standard English?

Reviewer #1: No

Reviewer #2: Yes

5. Review Comments to the Author

Reviewer #1: 1. Is the manuscript technically sound, and do the data support the conclusions?

--First, I would be cautious about drawing any conclusions--even negative conclusions-- from moderation analysis, as was done in Study 1, given the sample size. Roger Giner-Sorolla has a blog where he discusses in detail the specific problems of power analyses in moderation, and how GPower is not adequate.

Second, I would also be cautious about using difference scores, as was done for leadership behaviors. In some literatures this is an accepted practice, however there are issues with this--for one example, see Cafri, van den Berg, & Brannick, 2010. Would it be a better practice to include communal and agentic traits/behaviors as separate mediators? PROCESS accepts this. If not, why not?

Third, I did not understand the rationale for including Contrast 1 and Contrast 2 in the same analyses, nor how they were specifically related to Hypothesis 1.

Fourth, I did not understand the argument made on lines 972-986, specifically to how these results challenge explanations of the glass cliff. I didn't see anything in the Intro or the hypotheses that specifically opposed these explanations, if that is what was meant by "challenging." Or does this have do with using women to signal change? I would be wary about concluding that the "signalling change hypothesis" (to say) doesn't work. These results do not completely support it, but there is some support and that could be meaningful. On this point, however, I was not clear. I understood that Studies 2 and 3 assessed this "hypothesis" in different ways, but the implications of the conflicting results was not immediately clear. I think it could be worthwhile to set aside a small space in the discussion section to explicitly discuss this aspect.

Fifth, it seemed like the power analysis in Study 1 was introduced in between the treatment of Hypotheses 2 and 3. Did this also apply to Hypotheses 1 and 2? If they did, it was not immediately clear what the power analysis said about the results from H1 and H2, particularly because the power analysis was presented in terms of Odds Ratios but the results were presented in terms of betas.

Sixth, there are many references to the glass cliff but I do not recall seeing an explicit explanation of what the glass cliff actually is.

Seventh, I did not see a satisfactory explanation, or a potential explanation, for why there was a role for leadership behaviors but not leadership traits in Study 1. The explanation on lines 528-530 was not clear.

Eighth, I did not understand the point or implications of the analysis described on lines 502-504.

Ninth, I did not understand the reference to "competence ratings" or the consequent implications on lines 538-540.

Tenth and finally, I was not convinced by the rationale to exclude participants who chose the extreme agentic candidate in Study 1. Authoritarian, tough, and competitive would not describe my preferred candidate but others may prefer this style--in any case, I didn't think that it is an obviously wrong choice that should be excluded.

2. Has the statistical analysis been performed appropriately and rigorously?

--I don't have any specific issues besides things I've mentioned above, particularly with use of difference scores. Is this a common feature of this specific literature? I would feel more comfortable, given the problems in difference scores, if they were not used or, at least, a convincing argument was made for their necessity.

4. Is the manuscript presented in an intelligible fashion and written in standard English?

First, it was a bit difficult to read in places. I think it would benefit from having the hypotheses put into their own specific section. The hypotheses were not complex but at times I had trouble keeping track of them and how they related to the results.

Second, I think the manuscript could benefit from another pass of the language; there were numerous little passages throughout that, after awhile, made it difficult to read. For example, I had trouble with lines 132-133.

Third, in some places (e.g., 855-856), the means and standard deviations were separated rather than being paired. This was confusing and honestly just kind of weird. Is there a reason why they cannot be paired?

Despite my perhaps negative tone in places I thought that this was a fascinating paper that has a lot to add to the literature.

Reviewer #2: PONE-D-20-16233: Contextualizing the Think Crisis-Think Female Stereotype in Explaining the Glass Cliff

Gendered Traits, Gender, and Type of Crisis

I appreciated the different designs across studies (e.g., within and between subjects) to test the hypotheses multiple ways. These designs also helped to compensate for particular flaws in any one study (e.g., lack of manipulation checks and counterbalancing candidate profiles in Study 1) and replicate the effects, even testing them across different cultures. I provide a few comments/suggestions below.

When on pg. 8 the authors note that they are expanding on past research by investigating “gendered traits simultaneously with information on candidates’ gender” rather than in isolation, I assumed they would be looking at the interaction of these two factors. Yet, aside from Study 3’s nonsignificant interaction, no mention was made of this in terms of predictions or results. Is there a reason to predict that gender and gendered behavior would interact with each other? These data seem to have the capacity to test for this effect (that is the benefit of including both variables in the same study design) rather than looking at each individually.

It is obvious from the graphs that both agency and communion are relevant to these leadership contents, although differences between conditions are tested using a difference score. In most cases, these average difference scores were negative, which means that even in the relational crisis in Study 2 agency was rated higher than communion (although this difference is less than in the other conditions). Thus, I would caution the authors not to make claims that imply that only communal traits are important in certain contexts – it appears that both types of traits are important. The difference between the ratings varies, but that difference could have stemmed from changes in agentic traits as well as communal traits so a difference in the means isn’t that informative unless you can also see the graph to visually (but not statistically) see what ratings are changing across conditions. If the authors want to make claims about the absolute value of the agentic or communal demands of the position, they may want to rather test for difference within each trait separately. Otherwise, any summary of the results and conclusions should make it clear that it was the difference between the traits or leadership behaviors that varied across conditions (and that agency was almost always rated higher than communion).

I would have like more discussion of cultural differences and similarities across the studies, and what that means for choosing leaders in crisis situations, in the discussion section.

How were the communal and agentic leadership behaviors chosen? Is there evidence (e.g., outside ratings or manipulation checks) that these are agentic behaviors? They are noted as task-oriented behaviors, but don’t strike me as particularly agentic - delivering on a goal or meeting an objective would look different based on what the goal or objective is.

On pg. 38, now that the dependent variable has switched to one of three contexts, be careful how you talk about the results – saying that “female candidates were preferred by 53.7% of participants in the relational, 48.2% in the financial crisis, and 36.4% in the no crisis condition” doesn’t seem to make sense in this case, when there were no relational, financial, and crisis conditions but instead participants selected between these positions.

6. PLOS authors have the option to publish the peer review history of their article (what does this mean?). If published, this will include your full peer review and any attached files.

Reviewer #1: **Yes: **Conrad Baldner

Reviewer #2: No

---

## [Author Response · Author response to Decision Letter 0]

19 Sep 2020

Response to Reviewers

PONE-D-20-16233: Contextualizing the Think Crisis-Think Female Stereotype in Explaining the Glass Cliff Gendered Traits, Gender, and Type of Crisis

We tended closely to the instructions outlined in the two files and hope everything is fine now. (We have not highlighted the formatting as track-changes.)

2. Please include the full name of the Institutional Review Board that approved your study in the ethics statement.

We have added this information. However, the manuscript is not completely anonymous anymore as it states the affiliation of one of the authors.

E1. Problems with the data and analyses

E1a. Data: In Study 1, you excluded participants who chose the extreme agentic filler candidate. I disagree with this choice (also see comment from Reviewer 1). You cannot exclude participants just because they do not respond the way you expect them to unless there are sufficient reasons for doing so (e.g., those who do not pass the manipulation check). Thus, I would like you to put these participants back to your data set.

We have included the participants who chose the extreme agentic candidate in the analysis and updated the data-base on OSF. His choice was added to the choice of “Male” candidates and “agentic candidates”. The N is now 319 (previously 298). This information was changed in the abstract and method section. 

• Following the addition of these participants results have not changed. 

• Our analyses where we control for potential participant gender effects showed that participant gender does not produce any effects for the tests of H1 and H2 anymore. Thus, we deleted this section from the Supporting information section.

• Participant gender, however, had moderating effects on evaluations of the relevance of leadership behavior and traits which are fully reported in the Supporting information section.

We added the dataset including participants who chose the extreme agentic male to OSF.

E1.b Data analysis: In Study 1, you include two contrasts in the regression models. However, it is not clear to me how the Contrast 2 could be interpreted (also see comment from Reviewer 1). I assume that the inclusion of the Contrast 2 is to ascertain whether the two experimental conditions differ from the control condition but not in the way you expected (which was tested in Contrast 1). In this case, the contrast should be 1 = relational crisis, 0 = no crisis, 1 = financial crisis so that the intercept is more interpretable. In addition, the interaction term (gender traits x candidate sex) was not tested consistently across studies (see comment from Reviewer 2). Please test it across all studies. In addition, if the contrast 1 is significant, please indicate whether the two comparisons were both significant (relational crisis vs. no crisis; financial crisis vs. no crisis) or not. 

In Studies 1 and 2 we opted for a Contrasts analysis. Contrasts are widely recognized as a useful tool for testing precise predictions about differences between groups, which are derived from theoretical models (Brauer & McClelland, 2005; Judd & McClelland, 1989; Judd, McClelland, & Ryan, 2017). They are preferred to omnibus tests with multiple degrees of freedom (usually followed up by post hoc comparisons) because they offer a higher level of precision in hypothesis testing and also they have the advantage of greater statistical power. 

Regarding our hypotheses H1 and H2, we now detail the comparison tested by each contrast. For instance for H1, this part reads: 

“We conducted a logistic regression on the choice of candidates as a function of gendered traits (0 = agentic candidates, 1 = communal candidates). Following our prediction in H1, we entered two orthogonal contrasts for crisis type (Contrast 1: 1 = relational crisis, 0 = no crisis, -1 = financial crisis; Contrast 2: 1 = relational and financial crises, -2 = no crisis), and organizational role (-1 = leader; 1 = employee) as well as their interactions. The C1 contrast tests the difference between relational crisis versus financial crisis. The C2 contrast verifies whether the no crisis condition is situated in-between by testing the comparison between the no crisis versus relational and financial crises taken together. To support a linear effect, contrast C1 should be significant, but not C2.” (p19 lines 437-445)

Moreover, in order to be clearer in the report of our results, we adapted the description of Hypotheses 1 and 2, as well as the report of the linear effects in the Results by always saying that the condition coded as 1 is different from the condition coded as -1, and that the condition coded as 0 is situated in-between these conditions. 

Example: “We thus predict in Hypothesis 1 a linear effect: A communal (as opposed to agentic) leader will be preferred in a crisis context that contains communal problems (e.g., relational disharmony between employees) over a crisis scenario that contains agentic problems (e.g., financial problems), and a no crisis context will be situated in-between.” (p11, lines 240-244) 

And in the Results section: “Consistent with our main hypothesis H1, the C1 contrast showed that overall participants were more likely to select communal leaders in the relational crisis (50.00 %) than in the financial crisis (27.20 %), B = 0.49, χ2 (1, N = 319) = 12.15, p = .001, eB = 1.64 (Table 1), with the no crisis context (38.50 %) situated in-between (i.e., contrast C2 had no effect, B = 0.02, χ2 (1, N = 319) = 0.03, p = .861, eB = 1.02). ” (p19, lines 446-452)

Brauer, M., & McClelland, G. (2005). L’utilisation des contrastes dans l’analyse des données: Comment tester des hypothèses spécifiques dans la recherche en psychologie. [Using contrasts in data analysis: How to test specific hypotheses in psychology research]. L’Année Psychologique, 105, 273–305.

Judd, C. M., & McClelland, G. H. (1989). Data analysis: A model comparison approach. San Diego, CA: Harcourt Brace Jovanovich.

Judd, C. M., McClelland, G. H., & Ryan, C. S. (2017). Data Analysis: A Model Comparison Approach to Regression, ANOVA, and Beyond. Routledge (3rd ed.). Abingdon, UK: Routledge.

In addition, the interaction term (gender traits x candidate sex) was not tested consistently across studies (see comment from Reviewer 2). Please test it across all studies

Study 3 was specifically designed to allow for testing the interaction between gender traits x candidate sex on the choice of crisis type. This was possible because we reversed the study design that we used in the studies 1 and 2 such that Gender traits and Candidate sex became the independent variables and Crisis type the dependent variable. In Studies 1 and 2, the crisis type (relational versus financial versus no crisis) was the independent variable, and the Gender traits and Candidate sex the dependent variables. This design used in Studies 1 and 2 did not allow for the test of an interaction effect in the classical sense. The DV can only be Gendered traits or candidate sex but not the interaction of both.

E1.c Lastly, I agree with the Reviewer 1 that you should be cautious about using difference scores. It would be better to test them separately because communal and agency traits are of two dimensions rather than two poles of one dimension.

In the old version of the paper, we used difference scores as we were interested in the relative distance between the two concepts. This approach was also used in Kulich et al. (2018). However, in this article they may have used difference scores as in serial mediation one could not include agency and communion simultaneously. 

We agree with your argument that the literature shows that agency and communion are independent dimensions; also using difference scores does not reveal which one of the dimensions contributes more or less to the observed effects. Thus, we embraced your suggestion to use these measures separately. Following this decision we came to the conclusion that we can formulate a hypothesis on communion, but not for agency based on what we know from the literature.

We included the following changes:

1. Following this change in the analysis strategy, we adapted our hypothesis H3. We now frame it around changes in communal behavior and traits (instead of speaking of “communal versus agentic” attributions).

“Participants will ascribe higher relevance to communal leadership (i.e., behavior and traits) in a relational crisis scenario compared to a financial crisis, with the no-crisis context situated in-between. We expect this outcome to account for the linear preference effect of communal candidates predicted in H1.” (p12 lines 273-276)

2. Moreover, we added: “We did not make inverse predictions for agency, as the backlash literature on agentic female candidates shows that even if agency is considered relevant, lower relevance of agency does not necessarily lead to the choice of women [45]. In the context of glass cliff choices, we argue that the presence of communion-related attributions are likely to be decisive for the preference of communal leaders, rather than the absence of agentic ones. ” (p12 lines 277-282)

Descriptive results on the agency dimension can be found in the Supporting information.

E2. Problems with presentation

E2a. literature gaps: There are some choices made in the method section in which readers have not been informed (e.g., the organizational roles; the distinction of traits and behaviors).

We described the role of Organizational Roles under the subheading of “The present studies” p13, line 286. In order to make this section more visible we restructured the entire section and inserted a subheading: “Organizational role”. (p14, lines 303-329)

We added an explanation on the leadership measures in the “Measures” section in Study 1: 

“Models in the leadership literature consider both leadership traits and behaviors [17]. Traits are generally considered personality elements that are inherited or acquired through socialization and are often connected to gendered traits of identity that are viewed as more stable across different situations, traits such as being sensitive, empathetic, or kind. In contrast, behaviors or styles of leadership (e.g., transformational-transactional) are considered to be learned, assuming that one can be trained to flexibly use them across different situations (for a review comparing leadership effects across these dimensions see [56]). Because stereotypes concern expectations for how people are (or should be) and how they behave (or should) [57], we measured both traits and behaviors in an effort to capture a global picture of communal and agentic aspects of leadership.” (p18, lines 402-411)

E2b. Please reorder the information, move the sensitivity test to the method section under participants (because the information pertains to the sample size) and address the concern raised by Reviewer 1.

We moved the sensitivity power analysis to the method section in Study 1. Moreover, we have redone the analysis based on the new sample size of Study 1 (obtained after reincluding in the data set the participants who chose the extreme agentic candidate).

Regarding the concern raised by Reviewer 1 (“Fifth, it seemed like the power analysis in Study 1 was introduced in between the treatment of Hypotheses 2 and 3. Did this also apply to Hypotheses 1 and 2? If they did, it was not immediately clear what the power analysis said about the results from H1 and H2, particularly because the power analysis was presented in terms of Odds Ratios but the results were presented in terms of betas”) -- The sensitivity power analysis concerns H1 and H2. In the report of results for H1 and H2, we present both the betas and the Odd Ratios. In the results report, the odd ratios are noted as “eB “: (for example: “Consistent with our main hypothesis H1, the C1 contrast showed that overall participants were more likely to select communal leaders in the relational crisis (50.00 %) than in the financial crisis (27.20 %), B = 0.49, χ2 (1, N = 319) = 12.15, p = .001, eB = 1.64 (Table 1), with the no crisis context (38.50 %) situated in-between (i.e., contrast C2 had no effect, B = 0.02, χ2 (1, N = 319) = 0.03, p = .861, eB = 1.02). ”. (p19, lines 446-452)

E2c. I would rather see tables with all the results than the bar figures that contain very little information. Right now, we only get a glimpse of the logistic models, without seeing the whole picture.

We replaced Figures on the logistic regression results (Figures 1 and 4 became Tables 1 and 2). Figures on leadership traits and leadership behaviors were removed from the article but can be found as tables in the supporting information (Tables S1 and S2). Mediational analyses are represented in Figures 1 and 2.

E2d. please list the case number for the ethical approval in the text.

We added the ethic record numbers.

E2e. Whenever a main effect appears, please conduct post hoc comparisons to ascertain whether the two experimental conditions differ from the control condition (e.g., on page 30, only mean scores were reported without the p-values for the comparisons).

This comment refers to the point we discussed in our response to E1b. Since our hypotheses predict a specific pattern for the experimental conditions, namely a linear pattern, we used 

Linear Contrasts analyses with the two orthogonal contrasts C1 and C2 (which we thoroughly discuss in our response to the E1b comment). As mentioned in E1b, this approach is preferred to omnibus tests and post hoc comparisons because it offers greater statistical power. Moreover, in designing our studies, we conducted a priori power analyses for the tests of the main hypotheses based on a Contrasts analysis plan (and corresponding estimated effect sizes). 

E2f. I am very confused in terms of how the analysis was done done in Study 3. You stated that “We conducted multinominal logistic regressions...” on line 870 but listed “crisis type as outcome variable (1 = relational crisis, 2 = financial crisis, 3 = no crisis...” on lines 872-875. You further reported findings in the first regression model and stated that “the probability to choose the relational crisis (versus no crisis) on lines 876 – 877 and further “the probability to choose the no crisis (versus final crisis)... on line 879. I am not sure how categorical variables can be numbered and entered in a regression model and why the reference group kept changing in the findings of the same regression model. Findings in Study 3 should be reported so readers understand what was done and can interpret the findings themselves.

We revised the presentation of the analysis in Study 3, in order to better clarify the steps conducted in the multinomial logistic regression. We agree that ordering the categories of the outcome variable “(1 = relational crisis, 2 = financial crisis, 3 = no crisis... on lines 872-875.” may have been confusing for the reader since the multinomial logistic regression is used when the dependent variable is nominal (meaning that the categories cannot be treated as ranks). We now also state explicitly that we conducted two multinomial logistic regressions.

This paragraphs now reads: 

“We conducted two multinominal logistic regressions entering candidate gendered traits (-1 = agentic, 1 = communal candidates), candidate gender (-1 = male, 1 = female candidates), and their interaction as predictors, and crisis type as the outcome variable. In the first multinomial logistic regression, we specified the no crisis condition as the reference category. Thus, in this logistic regression model we were interested in the probabilities of choosing the relational crisis (versus no crisis), and, respectively, the financial crisis (versus no crisis), as a function of the predictors. In the second multinomial logistic regression, we specified the financial crisis condition as the reference category. This regression model informed us about the probability of choosing the relational crisis (versus financial crisis) and, respectively, the no crisis (versus financial crisis) situation, as a function of the predictors.” (p41, lines 934-944)

E2g. There are places that it’s quite difficult to follow/read. Please carefully read your manuscript and revise accordingly or seek editing services.

Reviewer #1:

1. Is the manuscript technically sound, and do the data support the conclusions?

R1.1.1 First, I would be cautious about drawing any conclusions--even negative conclusions-- from moderation analysis, as was done in Study 1, given the sample size. Roger Giner-Sorolla has a blog where he discusses in detail the specific problems of power analyses in moderation, and how GPower is not adequate.

Indeed, we agree that the sample size in Study 1 may not be big enough to adequately power the interaction effects involving the organisational role . We now acknowledge this limit when commenting on these effects (“Regarding our exploratory analysis of organizational role, no effect on the preference variables was shown but we cannot conclude that this variable does not have an impact, as our study did not have the power to test this effect decisively. ” p24, lines 548-550). Related to this issue, we specifically designed Study 2 to increase statistical power for the main effects that are of interest for us: “The main aim of this study was to replicate and extend the findings of Study 1 with a larger sample size based on a priori power analysis. Moreover, to increase power, we used a more simple design by fixing the participant organizational role to that of a leader (therefore not manipulating organizational role), and we provided only a list of four candidates to choose from (the fifth extreme agentic male candidate was not included).” (p25-26, lines 582-586)

R1.1.2 Second, I would also be cautious about using difference scores, as was done for leadership behaviors. In some literatures this is an accepted practice, however there are issues with this--for one example, see Cafri, van den Berg, & Brannick, 2010. Would it be a better practice to include communal and agentic traits/behaviors as separate mediators? PROCESS accepts this. If not, why not?

See our comment in E1c. 

R1.1.3 Third, I did not understand the rationale for including Contrast 1 and Contrast 2 in the same analyses, nor how they were specifically related to Hypothesis 1.

See our response to comment E1b. 

R1.1.4 Fourth, I did not understand the argument made on lines 972-986, specifically to how these results challenge explanations of the glass cliff. I didn't see anything in the Intro or the hypotheses that specifically opposed these explanations, if that is what was meant by "challenging." Or does this have do with using women to signal change? I would be wary about concluding that the "signalling change hypothesis" (to say) doesn't work. These results do not completely support it, but there is some support and that could be meaningful. On this point, however, I was not clear. I understood that Studies 2 and 3 assessed this "hypothesis" in different ways, but the implications of the conflicting results was not immediately clear. I think it could be worthwhile to set aside a small space in the discussion section to explicitly discuss this aspect.

This was indeed not what we meant. We replaced “challenge” by “add to explanations for the glass cliff.” :

“By investigating variations in crisis typology, our findings add to explanations for the glass cliff. Specifically, beyond extending past research that has identified motivating factors for the glass cliff which do not value women’s or communal competences [13,63], we investigated crisis contexts where stereotypically feminine qualities are perceived as an added value for leadership effectiveness.” (p45 lines 1046-1050)

We extended the reasoning on signaling change in Study 3 as follows:

“Participants were asked to evaluate candidates according to their potential to signaling change and then chose a suitable company context. Following this chronology, participants might not have viewed the potential to signal change in the context of the missions in the future workplace of the candidate. It may therefore be a better strategy to ask participants to match the candidate with a company, and then to outline their reasoning for their choice. “ p44-45, lines 1018-1024

And in the general discussion: “We tested this idea, with Study 2 showing an association of a motivation to signaling change with the choice of candidates in crisis contexts. However, this was true for both the choice of female and male candidates. In Study 3, we made a further attempt to establish a link to a signaling change explanation by asking participants to first rate candidates on their potential to signaling change, and then to attribute them to a company context. No conclusive results were revealed, probably due to the lack of meaning of the signaling idea if a candidate is not evaluated on the background of a specific (crisis-) mission. Future research should investigate whether what is understood by signaling change is understood in different ways for men and women. Moreover, a direct test of the impact of signaling change intentions could be performed by manipulating or priming participants with a mission to signaling change, with a consecutive measure of their leadership preferences in terms of gender. ” (p. 46 lines 1055-1066)

R1.1.5 Fifth, it seemed like the power analysis in Study 1 was introduced in between the treatment of Hypotheses 2 and 3. Did this also apply to Hypotheses 1 and 2? If they did, it was not immediately clear what the power analysis said about the results from H1 and H2, particularly because the power analysis was presented in terms of Odds Ratios but the results were presented in terms of betas.

This point was addressed in our response to Comment E2b.

R1.1.6 Sixth, there are many references to the glass cliff but I do not recall seeing an explicit explanation of what the glass cliff actually is.

We extended the explanation p3 (lines 58-65) which now reads: “The “glass” metaphor refers to the subtle discriminatory nature of this phenomenon as precarious appointments are more likely for minority groups. Moreover, these subtleties can obscure the reality of discrimination for a specific individual, as the phenomenon only becomes visible by considering several cases in aggregate. The “cliff” metaphor also relates to the fact that precarious positions likely expose occupants to more scrutiny and criticism, and higher levels of stress, in addition to the risk of failure, with serious consequences for their careers [12].”

R1.1.7 Seventh, I did not see a satisfactory explanation, or a potential explanation, for why there was a role for leadership behaviors but not leadership traits in Study 1. The explanation on lines 528-530 was not clear.

We added the following reasoning: 

“Unexpectedly, the type of crisis did not significantly affect the ratings of relevance for communal traits. Thus, H3 was not supported for the trait dimension. The reasons for this inconclusive result could be a lack of power. However, a theoretical explanation may be that a relational crisis could be solved by anyone who uses the right style of leadership, i.e. communal behavior. Learned communal behaviors may be perceived by participants as more flexible, capable of being exhibited by anyone, be they female or male when it is relevant, as in a relational crisis. In contrast, communal leadership traits may be deemed by participants as more likely inherited, attributed to internal characteristics, more stable, and therefore potentially more strongly associated to women than men, with a chosen leader less agile to adapt to changing situations.” (p.24-25, lines 556-566,)

R1.1.8 Eighth, I did not understand the point or implications of the analysis described on lines 502-504.

We only put this analysis to give a complete picture. But as this does not test any hypothesis we moved it from the paper to Supporting information.

R1.1.9 Ninth, I did not understand the reference to "competence ratings" or the consequent implications on lines 538-540.

This passage was indeed not clear. We corrected it to the following:

“Our present research suggests that participants focused on a candidate’s match with behavioral expectations (communal behavior) as a remedy for a relational crisis. Thus, communal behavior may be judged to be a learned “competence” to actually change the situation. Yet, communal leadership expectations may be only weakly or partially related to the choice of a female candidate.” (p25 lines 574-578)

R1.1.10 Tenth and finally, I was not convinced by the rationale to exclude participants who chose the extreme agentic candidate in Study 1. Authoritarian, tough, and competitive would not describe my preferred candidate but others may prefer this style--in any case, I didn't think that it is an obviously wrong choice that should be excluded.

As explained in E1a we now present analyses including this candidate. Results have not changed following this inclusion.

2. Has the statistical analysis been performed appropriately and rigorously?

--I don't have any specific issues besides things I've mentioned above, particularly with use of difference scores. Is this a common feature of this specific literature? I would feel more comfortable, given the problems in difference scores, if they were not used or, at least, a convincing argument was made for their necessity.

See E1c. We now present separate analyses for each dimension.

4. Is the manuscript presented in an intelligible fashion and written in standard English?

R1.4.1 First, it was a bit difficult to read in places. I think it would benefit from having the hypotheses put into their own specific section. The hypotheses were not complex but at times I had trouble keeping track of them and how they related to the results.

We added a section heading “Hypotheses” in line 228 and we placed H1 and H2 in the subsection “Leader preferences” (line 229) and H3 in “Mechanism” (line 267).

R1.4.2 Second, I think the manuscript could benefit from another pass of the language; there were numerous little passages throughout that, after awhile, made it difficult to read. For example, I had trouble with lines 132-133.

This was indeed not very clear, we rephrased the passage as followed: 

“Following this idea, an experiment by [13] (Study 1) investigated whether the male-manager association holds in a crisis situation, and demonstrated that while this link holds in a healthy company context, people associated feminine traits with leadership in a crisis context. Thus, an underlying motivation for the choice of a woman may be that her gender is associated with “communal” leadership deemed more effective in the given situation. The choice of a woman could thus be in the “best interest” of the ailing company’s functioning, with the chosen woman being expected to effectively change the situation due to her competencies. ” p6 lines 136-143

Moreover, an English native speaker who knows the domain of the glass cliff proofread the entire paper to strengthen clearness.

R1.4.3 Third, in some places (e.g., 855-856), the means and standard deviations were separated rather than being paired. This was confusing and honestly just kind of weird. Is there a reason why they cannot be paired?

We corrected this and put them in pairs.

Despite my perhaps negative tone in places I thought that this was a fascinating paper that has a lot to add to the literature.

We thank you for your thorough reading and relevant comments to our work which hopefully helped us to produce a clearer version of our manuscript.

Reviewer #2:

I appreciated the different designs across studies (e.g., within and between subjects) to test the hypotheses multiple ways. These designs also helped to compensate for particular flaws in any one study (e.g., lack of manipulation checks and counterbalancing candidate profiles in Study 1) and replicate the effects, even testing them across different cultures. I provide a few comments/suggestions below.

R2.1. When on pg. 8 the authors note that they are expanding on past research by investigating “gendered traits simultaneously with information on candidates’ gender” rather than in isolation, I assumed they would be looking at the interaction of these two factors. Yet, aside from Study 3’s nonsignificant interaction, no mention was made of this in terms of predictions or results. Is there a reason to predict that gender and gendered behavior would interact with each other? These data seem to have the capacity to test for this effect (that is the benefit of including both variables in the same study design) rather than looking at each individually.

We did not expect any variations in preferences of different combinations of candidate traits and gender across the different crisis types. So for example, we do not know a reason to believe that communal men should face more backlash in one crisis condition than in another. Thus, we did not formulate any hypotheses in this sense. Having all in the same design allowed us to control for different combinations of traits and gender. The only way of testing such differences would be to oppose the choice of single candidates, e.g. compare the agentic with the communal male, but then the analysis would only consider one part of the design which will bias the estimates of the effects. We opted for the design of Study 3 to have one direct exploration of the potential effect of an interaction. Future research could more specifically focus on an interaction hypothesis.

R2.2. It is obvious from the graphs that both agency and communion are relevant to these leadership contents, although differences between conditions are tested using a difference score. In most cases, these average difference scores were negative, which means that even in the relational crisis in Study 2 agency was rated higher than communion (although this difference is less than in the other conditions). Thus, I would caution the authors not to make claims that imply that only communal traits are important in certain contexts – it appears that both types of traits are important. The difference between the ratings varies, but that difference could have stemmed from changes in agentic traits as well as communal traits so a difference in the means isn’t that informative unless you can also see the graph to visually (but not statistically) see what ratings are changing across conditions. If the authors want to make claims about the absolute value of the agentic or communal demands of the position, they may want to rather test for difference within each trait separately. Otherwise, any summary of the results and conclusions should make it clear that it was the difference between the traits or leadership behaviors that varied across conditions (and that agency was almost always rated higher than communion).

This is an important point. In order to be clearer, we report the two dimensions separately, communion in the main paper and agency in the Supporting information. The reason why we only focus on communion is explained in our comments to E1c.

R2.3. I would have like more discussion of cultural differences and similarities across the studies, and what that means for choosing leaders in crisis situations, in the discussion section.

We only dared to advance one point touching on potential cultural differences. As we have also a variation in sample type (students versus workers), we do not wish to go much further.

“It is of interest to note that despite a stronger preference for female CEOs in crisis than no crisis contexts, overall male CEOs were preferred in Study 1 in a Spain sample. In Study 2 with a US sample, no main effect of gender was observed, suggesting that in this context, female leadership has potentially become a more accessible, or acceptable concept. In the corporate world, relatively few women are found in top leadership positions, and particularly in Spain, management is still perceived as less compatible with female gender compared to other European countries and the United States of America [65].” (p 47, lines 1067-1082)

R2.4. How were the communal and agentic leadership behaviors chosen? Is there evidence (e.g., outside ratings or manipulation checks) that these are agentic behaviors? They are noted as task-oriented behaviors, but don’t strike me as particularly agentic - delivering on a goal or meeting an objective would look different based on what the goal or objective is.

Following the treatment of communion and agency as two independent measures (instead of difference scores) we had to reformulate our Hypotheses and came to the conclusion that we cannot make clear predictions for agency. Thus, only communion is considered in the present article. In future studies, we will reconsider your comment as concerns the content and interpretation of the agentic behavioral dimensions.

R2.5 On pg. 38, now that the dependent variable has switched to one of three contexts, be careful how you talk about the results – saying that “female candidates were preferred by 53.7% of participants in the relational, 48.2% in the financial crisis, and 36.4% in the no crisis condition” doesn’t seem to make sense in this case, when there were no relational, financial, and crisis conditions but instead participants selected between these positions.

We agree with your comment that given the design of Study 3 and the analyses that we conducted, presenting those percentages may be confusing and not particularly relevant. We inserted a Table 3 (p 42) in which we present all the percentages for crisis type choice as a function of candidate gendered traits and gender.

We thank you for your thorough reading and relevant comments to our work which hopefully helped us to produce a clearer version of our manuscript.

---

## [Decision Letter · Decision Letter 1]

9 Nov 2020

PONE-D-20-16233R1

Contextualizing the think crisis-think female stereotype in explaining the glass cliff:

Gendered traits, gender, and type of crisis

PLOS ONE

Dear Dr. Kulich,

Thank you for submitting your manuscript to PLOS ONE. After careful consideration, we feel that it has merit but does not fully meet PLOS ONE’s publication criteria as it currently stands. Therefore, we invite you to submit a revised version of the manuscript that addresses the points raised during the review process.

I have received comments from two previous reviewers and I share their view that the revised manuscript is very much improved. We, however, have some suggestions for the authors and would hope that the authors could incorporate these suggestions in the manuscript. In addition to the reviewers’ suggestions (also see the attached file), here are my suggestions.

I would suggest taking out the term “linear” in the hypotheses and text. The authors were comparing three conditions which are categorical. Thus, to suggest that the differences between conditions were linear seem a distraction to me without evidence that could be used to substantiate the claim. For example, in testing H1, the percentages of participants to select communal leaders the three conditions are 50% (relational crisis), 38.5% (no crisis), and 27.2% (financial crisis). The differences between the conditions were not equivalent so I am not sure how you could say that this effect is linear (I suspect that the authors use the linear effects implicit assumption in regression models and treat the evidence as if it is actually linear). If the authors disagree, please provide results that compare linear effects with other types effects (e.g., curvilinear, cube, etc.).Please include post-hoc comparison results in the tables so readers could read from the tables whether the numbers reported between either two conditions were significantly different.Please help readers understand why the roles expectations are important (imagine themselves to be leaders or employees) in the investigation of the glass cliff phenomenon and how this variation may have affect the hypothesized effects (or not). Are the candidates all expected to fill a leader’s role?Fig 1 on p. 22 was missing (only figure caption shown)Please revised the first sentence under the mediation analysis on p. 22. Shouldn’t it be communal behaviors rather than communal traits as the gendered traits did not have expected pattern in the above section.In studies 2 & 3, please specify the design before going details regarding the procedures. I am confused because the company performance (poor or strong) did not seem to be considered in the design.The information with regards to how participants got the manipulation check items wrong on p. 40 could be moved to the participant section when you specify who were excluded.Please provide the manipulation materials in the appendix.

We look forward to receiving your revised manuscript.

Kind regards,

I-Ching Lee

Academic Editor

PLOS ONE

Reviewers' comments:

Reviewer's Responses to Questions

**Comments to the Author**

1. If the authors have adequately addressed your comments raised in a previous round of review and you feel that this manuscript is now acceptable for publication, you may indicate that here to bypass the “Comments to the Author” section, enter your conflict of interest statement in the “Confidential to Editor” section, and submit your "Accept" recommendation.

Reviewer #1: (No Response)

Reviewer #2: All comments have been addressed

2. Is the manuscript technically sound, and do the data support the conclusions?

Reviewer #1: Yes

Reviewer #2: Yes

3. Has the statistical analysis been performed appropriately and rigorously? 

Reviewer #1: Yes

Reviewer #2: I Don't Know

4. Have the authors made all data underlying the findings in their manuscript fully available?

Reviewer #1: Yes

Reviewer #2: Yes

5. Is the manuscript presented in an intelligible fashion and written in standard English?

Reviewer #1: Yes

Reviewer #2: Yes

6. Review Comments to the Author

Reviewer #1: (No Response)

Reviewer #2: The authors have addressed the comments from the first version of the manuscript. I have only a couple of minor points:

If the odds ratio is noted at "eb", I would suggest the authors indicate this somehow as it is otherwise difficult to compare the power analysis to the results.

Confidence intervals were not consistently given for all results.

7. PLOS authors have the option to publish the peer review history of their article (what does this mean?). If published, this will include your full peer review and any attached files.

Reviewer #1: **Yes: **Conrad Baldner

Reviewer #2: No

---

## [Author Response · Author response to Decision Letter 1]

19 Nov 2020

EDITOR COMMENTS

1. I would suggest taking out the term “linear” in the hypotheses and text. The authors were comparing three conditions which are categorical. Thus, to suggest that the differences between conditions were linear seem a distraction to me without evidence that could be used to substantiate the claim. For example, in testing H1, the percentages of participants to select communal leaders the three conditions are 50% (relational crisis), 38.5% (no crisis), and 27.2% (financial crisis). The differences between the conditions were not equivalent so I am not sure how you could say that this effect is linear (I suspect that the authors use the linear effects implicit assumption in regression models and treat the evidence as if it is actually linear). If the authors disagree, please provide results that compare linear effects with other types effects (e.g., curvilinear, cube, etc.).

We can see your point, and we agree that the term “linear” may lead to confusion as we refer to three categorical conditions. In the literature theorising the Contrast analysis approach, a linear trend among three conditions does not actually make the claim that a straight line fits perfectly the relation between the independent and the dependent variables. A linear trend is tested by considering together the two orthogonal contrasts, in our case, the C1 and C2. The C1 alone tests only the difference between the two extreme conditions: for instance, between relational crisis versus financial crisis. The C2 (which is called the “quadratic trend”), informs us whether the condition in-between is positioned in such a way as to more closely represent a linear trend as opposed to a quadratic trend. In order to be able to conclude that the three conditions depict a linear trend, the C1 should be significant and the C2 should not. The C2 not being significant means that the in-between condition is not significantly distanced from the middle point between the extreme conditions, hence a trend that is close to a linear one. If C2 was significant, this would imply that the in-between condition was closer to one of the extreme conditions, meaning that the differences between the conditions are not equivalent. However, we recognise that for a reader not familiar with this approach and terminology, the term linear will be misleading. We thus followed your recommendation and deleted, or replaced, the term “linear” in the entire manuscript (e.g. “In Hypothesis 2, which concerns gender, we thus also predicted a progressive pattern” (p12, line 263) ; “For communal leadership behavior, path ‘a’ showed a positive incremental effect of financial crisis - no crisis - relational crisis (C1) on communal behavior “ (p23, line 520)). We hope that this strategy has clarified our reports.

2. Please include post-hoc comparison results in the tables so readers could read from the tables whether the numbers reported between either two conditions were significantly different.

We do not report post-hoc comparisons in the article as this does not correspond to our analysis plan. The decision to use the Contrasts analysis approach was guided by two main points: the expected trend among our three conditions and the increased statistical power offered by contrast testing as opposed to omnibus tests and post-hoc comparisons. Thus, we powered our studies (in terms of sample sizes) based on this analysis plan. This means that any post-hoc comparisons will be underpowered. However, we do understand that the reader may be interested in having these analyses as well. We thus now provide two tables with post-doc comparisons in the Supporting information (Tables S1 Table and S3 Table).

3. Please help readers understand why the roles expectations are important (imagine themselves to be leaders or employees) in the investigation of the glass cliff phenomenon and how this variation may have affect the hypothesized effects (or not). Are the candidates all expected to fill a leader’s role?

Candidates in all three studies were expected to fill a leadership role. 

As concerns participants, in Study 1 we manipulated their role, by asking them to think about themselves as a leader or an employee. In Studies 2 and 3 participants, who were employed themselves in real life, were asked to put themselves in the shoes of someone who needs to do the best for the company in terms of focusing on appropriate functioning. The exact formulation was: “You will then be asked to evaluate these candidates and choose the one who is best suited to ensure appropriate functioning of the company.” and then “Please select the candidate whom you consider to be the most suitable for this position at Jefferson's.” We believe that this instruction made people act in the interest of the company and not in their own interest (be they leaders or employees in real life). 

We believe that glass cliffs should occur when people are being put in a decision maker’s position, which we now clearly state in the paper:

“As the recruitment for leadership positions is usually done by decision makers with managing functions in companies, rather than employees, the angle of the decision maker perspective is more likely to be source of glass cliffs decisions. Moreover, the potential motivations for glass cliff decisions discussed here also take the perspective of those in charge of hiring. Thus, in the present research we asked participants to focus primarily on this role.” (p14, lines 320-325)

As we argue on pages 13-14, employees may have distinct motivations. However, we did not make specific hypotheses concerning the choice of a leader. This part was very explorative. 

4. Fig 1 on p. 22 was missing (only figure caption shown)

We added both figures in the manuscript. (Sorry, we thought we only needed to upload them as separate files.)

5. Please revised the first sentence under the mediation analysis on p. 22. Shouldn’t it be communal behaviors rather than communal traits as the gendered traits did not have expected pattern in the above section.

Thank you for pointing out this error. We have replaced traits by behavior.

6. In studies 2 & 3, please specify the design before going details regarding the procedures. I am confused because the company performance (poor or strong) did not seem to be considered in the design.

We moved the study design to the top of the Procedure sections for both Studies 2 (p28, line 633) and 3 (p40, line 904). 

The no crisis condition is the “strong” and the two crisis conditions are the “poor” conditions and we realize that this has not been made explicitly clear to the reader. We now write in Study 2 “Company performance was presented as either poor or as strong (hereafter between brackets). Poor performance was presented for the two crisis conditions and strong performance for the no crisis condition.” (p28, lines 638-640)

7. The information with regards to how participants got the manipulation check items wrong on p. 40 could be moved to the participant section when you specify who were excluded.

Thank you. Indeed this part fits better in the participant section to which we have moved this part.

8. Please provide the manipulation materials in the appendix.

We created an appendix with all manipulation materials. 

Please note: Doing an exact translation of the Spanish materials from Study 1 I got aware of some imprecisions which I corrected in the manuscript’s method section of Study 1. 

Reviewers' comments:

R2.1 If the odds ratio is noted at "eb", I would suggest the authors indicate this somehow as it is otherwise difficult to compare the power analysis to the results.

Confidence intervals were not consistently given for all results.

We added a note in the article indicating that “eb” represents the odd ratio (line 455). 

We added CIs to all logistic regressions in Studies 1 and 2. 

R1.1. Lines 394-395:

The authors wrote: "He was described with extreme, unmitigated agentic traits that included negative content (e.g., authoritarian, tough, competitive; see [55]), aiming to disqualify him by this unsympathetic description."

I'm not sure if the Helgeson (1994) reference cited here is very relevant. It talks about agency and communion in men and women and how it relates to their own well-being, and not how they are perceived by others, if I understood it correctly. Granted this was the least popular category, but I'm not sure if this is the best reference to support this.

Thank you for spotting this. We replaced the reference by Abele & Wojciszke (2007).

R1.2. Line 464:

The authors wrote: "The in H2 expected linear effect of..." Typo? 

We rephrased as follows: “The effect of crisis type C1 that was expected in H2,...” (p21, line 471)

R1.3. Lines 562-566: "In contrast, communal leadership traits may be deemed by participants as more likely inherited, attributed to internal characteristics, more stable, and therefore potentially more strongly associated to women than men, with a chosen leader less agile to adapt to changing situations.”

I'm wary about the perception that these traits are inherited. We speak about psychological traits, but would people think that something like communality is literally inherited? I think that it would be enough to say that communal traits are likely perceived to be internal and stable.

Thank you for spotting this imprecision. We have rephrased this sentence as follows: 

“In contrast, communal leadership traits may be attributed by participants to internal characteristics and considered as more stable, and therefore potentially more strongly associated to women than men, with a chosen leader less agile to adapt to changing situations. ” (p25, line 571)

R1.4. Lines 776-779: The authors wrote: "For leadership behavior, path ‘a’ showed that the linear effect of relational crisis versus no crisis versus financial crisis (C1) was related to more communal behavior (B = 0.40, SE = 0.08, p < .001, 95% CI [0.23, 0.56]), and more communal traits (B = 0.21, SE = 0.06, p < .001, 95% CI [0.09, 0.34]." 

Would it be more correct to refer to preference for, or relevance of, communal behavior/traits, instead of "more" communal behavior/traits?

Thank you, we have corrected it to “For leadership behavior, path ‘a’ showed that the incremental effect of financial crisis - no crisis - relational crisis (C1) was related to higher relevance ratings of communal behavior (…)” (p35, 787) 

R1.5. Lines 805-815: The authors didn't find support for a mediating effect on the relationship between crisis type and candidate gender preference. I had two comments. First, is it possible to see the actual results for the mediation, above knowing that it was not significant? Second, would it have been consistent with the authors' hypotheses to test for the mediating role of signalling change on the relationship between crisis type and preference for communal vs. agentic leader?

First: We have added the statistical indicators. “The mediation effect was not significant (B = - 0.01, SE = 0.03, 95% CI [-0.08, 0.06]). Thus, although signaling change was associated more with candidates chosen in both crisis contexts (path ‘a’: B = 0.36, SE = 0.04, p < .001, 95% CI [0.27, 0.44]), it was not specifically more associated with the choice of a woman (path ‘b’: B = -0.03, SE = 0.09, p = .775, 95% CI [-0.21, 0.16]).” (p36-37)

Second: No in the glass cliff literature the signaling hypothesis concerns the visible replacement of a male by a female (or a White by a Non-White) leader. When we look at trait quality this points more towards the actual qualities of a person, their way of leading, and this would be predicted to be explained by the relevance ratings of leader qualities (see mediation analysis in the paper).

We describe in this letter results if mediation was tested for signaling change for the candidate choice in terms of their gendered traits: The mediation effect was not significant (B = - 0.03, SE = 0.02, 95% CI [-0.07, 0.006]). Signaling change was associated more with candidates chosen in the relational than no crisis context (path ‘a’: effect of C1 B = -0.20, SE = 0.04, p < .001, 95% CI [-0.28, -0.12]; C2 which compares the no crisis to the two crisis contexts was also significant: B = -0.51, SE = 0.04, p < .001, 95% CI [-0.65, -0.37]). Signaling change was not associated with the choice of a communal candidate (path ‘b’: B = 0.16, SE = 0.10, p = .119, 95% CI [-0.04, 0.36]).”

R1.6. Lines 819-822:The authors wrote: "However, while the financial crisis was the least preferred condition for communal candidates (consistent with H1), the no crisis condition was the least preferred for female candidates (consistent with H2)."

I don't have an issue with the info presented here, but I though that the wording was a bit awkward. In particular, I suspected following "However, while..." that the authors would present some contradictory findings, instead of stating that there hypotheses were supported.

We did not intend to express contradiction, so we aligned it with your suggestion: “As expected, while the financial crisis was the least preferred condition for communal candidates (consistent with H1), the no crisis condition was the least preferred for female candidates (consistent with H2).” (p37, line 832)

R1.7. Lines 990-996: It wasn't entirely clear what was being analyzed here. In particular, I had trouble understanding "an organizational context for which the importance of the communal leadership behavior was stronger compared to the match for agentic candidates". Is this the relational context?

We rephrased the sentence to make it clearer. This part now reads: 

“The perceived relevance of the communal behavior was higher for the type of crisis matched to the communal candidate (M= 5.82, SE= 0.10) compared to the one matched to the agentic candidate (M = 5.24, SE = 0.10).” (p45, line 1006)

---

## [Decision Letter · Decision Letter 2]

22 Jan 2021

Contextualizing the think crisis-think female stereotype in explaining the glass cliff:

Gendered traits, gender, and type of crisis

PONE-D-20-16233R2

Dear Dr. Kulich,

We’re pleased to inform you that your manuscript has been judged scientifically suitable for publication and will be formally accepted for publication once it meets all outstanding technical requirements.

Kind regards,

I-Ching Lee

Academic Editor

PLOS ONE

Additional Editor Comments (optional):

Reviewers' comments:

Reviewer's Responses to Questions

**Comments to the Author**

1. If the authors have adequately addressed your comments raised in a previous round of review and you feel that this manuscript is now acceptable for publication, you may indicate that here to bypass the “Comments to the Author” section, enter your conflict of interest statement in the “Confidential to Editor” section, and submit your "Accept" recommendation.

Reviewer #2: All comments have been addressed

2. Is the manuscript technically sound, and do the data support the conclusions?

Reviewer #2: Yes

3. Has the statistical analysis been performed appropriately and rigorously? 

Reviewer #2: I Don't Know

4. Have the authors made all data underlying the findings in their manuscript fully available?

Reviewer #2: Yes

5. Is the manuscript presented in an intelligible fashion and written in standard English?

Reviewer #2: Yes

6. Review Comments to the Author

Reviewer #2: (No Response)

7. PLOS authors have the option to publish the peer review history of their article (what does this mean?). If published, this will include your full peer review and any attached files.

Reviewer #2: No

---

## [Editor Report · Acceptance letter]

19 Feb 2021

PONE-D-20-16233R2 

Contextualizing the think crisis-think female stereotype in explaining the glass cliff:  Gendered traits, gender, and type of crisis 

Dear Dr. Kulich:

I'm pleased to inform you that your manuscript has been deemed suitable for publication in PLOS ONE. Congratulations! Your manuscript is now with our production department. 

Kind regards, 

on behalf of

Dr. I-Ching Lee 

Academic Editor

PLOS ONE